# *Drosophila* ß<sub>Heavy</sub>-Spectrin is required in polarized ensheathing glia that form a diffusion-barrier around the neuropil

Nicole Pogodalla[1], Holger Kranenburg[1], Simone Rey[1], Silke Rodrigues[1], Albert Cardona [2,3,4] & Christian Klämbt [1✉]

In the central nervous system (CNS), functional tasks are often allocated to distinct compartments. This is also evident in the *Drosophila* CNS where synapses and dendrites are clustered in distinct neuropil regions. The neuropil is separated from neuronal cell bodies by ensheathing glia, which as we show using dye injection experiments, contribute to the formation of an internal diffusion barrier. We find that ensheathing glia are polarized with a basolateral plasma membrane rich in phosphatidylinositol-(3,4,5)-triphosphate ($PIP_3$) and the $Na^+/K^+$-ATPase Nervana2 (Nrv2) that abuts an extracellular matrix formed at neuropil-cortex interface. The apical plasma membrane is facing the neuropil and is rich in phosphatidylinositol-(4,5)-bisphosphate ($PIP_2$) that is supported by a sub-membranous ß<sub>Heavy</sub>-Spectrin cytoskeleton. *ß<sub>Heavy</sub>-spectrin* mutant larvae affect ensheathing glial cell polarity with delocalized $PIP_2$ and Nrv2 and exhibit an abnormal locomotion which is similarly shown by ensheathing glia ablated larvae. Thus, polarized glia compartmentalizes the brain and is essential for proper nervous system function.

[1] Institut für Neuro- und Verhaltensbiologie, Universität Münster, Münster, Germany. [2] MRC Laboratory of Molecular Biology, Cambridge Biomedical Campus, Cambridge, CB2 OQH, UK. [3] Physiology, Development and Neuroscience Department, University of Cambridge, Cambridge, UK. [4] HHMI Janelia Research Campus, Ashburn, VA, USA. ✉email: klaembt@uni-muenster.de

The complexity of the nervous system is not only defined by intricate neuronal networks but is also reflected by the cellular diversity of the different glial cells found throughout the nervous system. The number of functions attributed to the different glial cells is steadily growing and ranges from guiding and instructing functions during early brain development to synapse formation and plasticity, and general ion and metabolite homeostasis[1,2].

One of the key functions of glial cells lies in the establishment of boundaries that help compartmentalizing neuronal computing. A very tight boundary is the blood–brain barrier. In all invertebrates as well as in primitive vertebrates this barrier is established by glial cells, whereas higher vertebrates transferred this function to endothelial cells[3–5]. Smaller signaling compartments are established by astrocytes that tile the synaptic areas in the brain[6–8]. Oligodendrocytes insulate single segments of up to 20 different axons by forming myelin sheets around them[9]. Moreover, oligodendrocyte progenitor cells act as innate immune cells and participate in scar formation separating intact brain tissue from the injured site[10].

Invertebrates have a less complex nervous system and glial cells are clearly outnumbered by neurons. In the ant *Harpegnathos saltator* 27% of all brain cells are of glial nature, whereas in the fruit fly *Drosophila melanogaster* just 10% of all neural cells are of glial nature[11,12]. As in primitive vertebrates, glial cells form the blood–brain barrier and organize the entire metabolite traffic into and out of the nervous system and also integrate external stimuli such as circadian rhythms with general brain functions[13–16]. All neuronal cell bodies are situated beneath the blood–brain barrier and are surrounded by cortex glial cells. These cells regulate neuroblast division, nurture neurons, and phagocytose dying neurons[17–24].

Fly neurons project their dendrites and axons into the central neuropil. This morphologically distinct structure is separated from all neuronal cell bodies by the neuropil-associated glia, which comprises astrocyte-like glial cells and ensheathing glial cells[25–29]. Astrocyte-like cells form numerous fine and highly branched processes that infiltrate the neuropil. As in vertebrates they tile the synaptic neuropil into discrete units. They express a number of neurotransmitter transporters to clear synaptic spillover although they do not form frequent intimate contacts with individual synapses and in addition are able to secrete gliotransmitters[29–31].

The neuropil encasing ensheathing glia comprises two distinct classes that can be classified by morphological as well as molecular criteria[11,27,28,32]. In the larval nervous system, just four ensheathing glial cells are formed in each hemineuromer. Two of them embrace the neuropil and do not wrap around individual axons. In contrast, the other two larval ensheathing glial cells found in each abdominal hemineuromere exhibit a more complex morphology. In part they encase the neuropil as the other two ensheathing glial cells but in addition they also enwrap axons between the CNS/PNS boundary and the neuropil. They are thus called ensheathing/wrapping glial cells[27,28]. In the larva, the organization of the neuropil is relatively simple, but the adult CNS shows complex regional compartmentalization[33,34]. This is reflected by an increase in the number of ensheathing glial cells that are generated during pupal stages and, as in larval stages, fall into two morphological classes[32,35].

Several studies have already shed some light on the functional roles of ensheathing glia in the fly nervous system. First, it was demonstrated that ensheathing glial cells remove neuronal debris after injury utilizing the Draper pathway[24,36,37]. In addition to this immune and surveillance function, ensheathing glia can participate in neuronal signaling. Mutant analysis indicates the sulfite oxidase Shopper needs to be expressed by ensheathing glia

to regulate glutamate homeostasis in the neuropil[28]. Likewise, the Excitatory amino acid transporter 2 (Eaat2) functions in ensheathing glia to modulate sleep in the adult[38]. In addition, cell type specific knockdown experiments demonstrate that the voltage-gated potassium channel encoded by the gene *seizure* (*sei*) is required in ensheathing glia to protect flies from acute heat-induced seizures[39].

Given the apparent role of ensheathing glia in forming neuronal compartments and modulating their function, we initiated a comprehensive analysis of this—as we found—highly polarized cell type. The plasma membrane of the ensheathing glia facing the neuropil is rich in the phospholipid $PIP_2$ and is supported by submembranous $ß_{Heavy}$-Spectrin. Thus, it can be assumed that the apical cell domain is oriented towards the neuropil. The basolateral plasma membrane exhibits an accumulation of the $Na^+/K^+$ ATPase Nervana 2 (Nrv2) suggesting that sodium ion dependent transport is polarized in these cells. In addition, basally localized Integrins anchor the ensheathing glia to a specific extracellular matrix (ECM) formed at the interface of neuropil and CNS cortex. To study the functional relevance of ensheathing glia, we ablated these cells using a split Gal4 driver that we had established. The absence of ensheathing glia triggers a compensatory growth of astrocyte-like cells which, however, does not completely restore an internal diffusion barrier normally promoted by ensheathing glia. Animals lacking ensheathing glia show abnormal larval locomotion and reduced longevity. Reduction of $ß_{Heavy}$-Spectrin expression affects ensheathing glia cell morphology and leads to mislocalization of $PIP_2$. Moreover, larvae with ensheathing glia specific $ß_{Heavy}$-*spectrin* knockdown or larvae lacking ensheathing glia show similar abnormal locomotor phenotypes as $ß_{Heavy}$-*spectrin* mutant larvae. In conclusion, we show that separation of the neuropil by polarized ensheathing glia is required for nervous system function.

## Results

**Ensheathing glia encase the neuropil throughout development.** Four ensheathing glial cells are formed in each hemineuromere during mid embryogenesis and are associated with the neuropil[27]. Two of these cells only encase the neuropil, whereas the two other also wrap axons connecting the peripheral nerves with the neuropil (ensheathing/wrapping glial cells, Fig. 1)[27,32,40]. To target the larval ensheathing glial cells, we compared the activity of the previously used Gal4 lines (*NP6520-Gal4, 56F03-Gal4, nrv2-Gal4,* and *83E12-Gal4*) and found the *83E12-Gal4* driver as the most specific one for both larval and adult CNS[27,28,32,41]. Cell counts and 3D reconstructions (see below) suggest that *83E12-Gal4* is active in all larval ensheathing glial cells. However, since no 3D reconstruction has been conducted for the adult TEM volume we cannot exclude that some *83E12-Gal4* negative ensheathing glial cells exist.

During embryonic stages, the ensheathing glia initially cover the dorsal domain of the neuropil (Fig. 1a, a′). In first instar larvae, thin processes of the ensheathing glia begin to encase the neuropil (Fig. 1b, b′). Notably, small gaps between individual ensheathing glial processes are still detectable in the second instar larva (Fig. 1c, c′). In the third larval instar, ensheathing glia appear to form a closed case around the neuropil. The ensheathing glia also send processes around the dorsally located neuronal cell bodies (Fig. 1d′, Supplementary Fig. S1). Multi-color flipout (MCFO) labelling demonstrates that ensheathing glia processes tile the neuropil (Fig. 1e–g, j) to separate it from the CNS cortex. The cortex glia can be visualized using the Gal4 drivers *55B12-Gal4; NP2222-Gal4* and *NP0577-Gal4*[28,42]. These drivers showed that no cortex glial cells reside at the dorsal surface of the ventral nerve cord (Fig. 1h–j). This notion is further

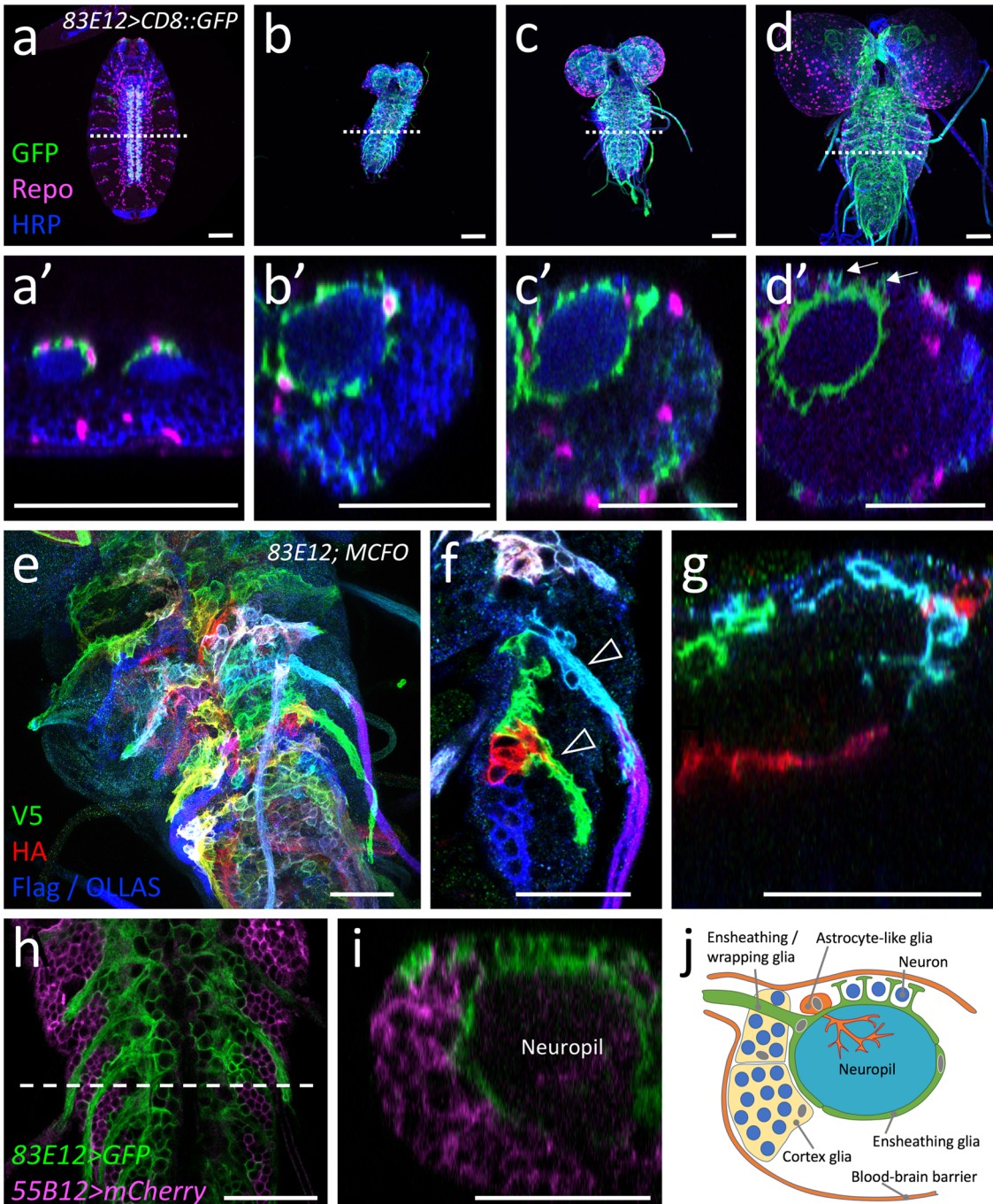

**Fig. 1 Development of ensheathing glia.** Representative images are shown. **a–d** Dissected larval CNS of increasing age with the genotype [*83E12-Gal4, UAS-CD8::GFP*], stained for GFP (green), Repo (magenta) and neuronal membranes (anti-HRP, blue), anterior is up. The positions of the orthogonal section shown in **a′–d′** is indicated by a dashed line. **a**, **a′** In a stage 16 embryo ensheathing glial cells have not yet covered the neuropil (arrowheads). **b**, **b′** First instar larval CNS. **c**, **c′** Second instar larval CNS. **d**, **d′** CNS of a wandering third instar larva. The arrows point towards dorsal protrusions of the ensheathing glia engulfing dorsal neurons. **e** MCFO labeling of ensheathing glia in third instar larva stained for the expression of V5 (green), HA (red), and FLAG and OLLAS epitopes (blue). *flp* expression was induced for one hour during first instar larval stage. Note that ensheathing glia tile the ventral nerve cord. **f** Two distinct ensheathing/wrapping glia cells cover the nerve root and part of the neuropil (arrowheads). **g** Ensheathing glia occupy specific territories in the neuropil. **h**, **i** Third instar larval nerve cord with the genotype [*55B12-Gal4, 83E12-LexA, UAS-CD8::mCherry, LexAop-GFP*]. All cortex glia cells are labelled by mCherry expression (magenta). Ensheathing glial cells are labelled by GFP expression (green). The dashed line indicates the position of the orthogonal view shown in (**i**). **j** Schematic view on a cross section through a hemineuromere indicating the position of the different glial cells. Astrocyte-like cells and ensheathing glial cells localize close to the neuropil and the axons connecting the neuropil with the periphery. The cortex glia covers lateral and ventral neuronal cell bodies. Scale bars **a–i** are 50 µm.

corroborated by split-GFP experiments where GFP is reconstituted at the baso-lateral boundary between cortex and neuropil but not at the dorsal surface of the ventral nerve cord (Supplementary Fig. S1a–f'). Here, several large glutamatergic neurons are found that appear to be encased by processes of the ensheathing glia (Supplementary Fig. S1g–l). In conclusion, the ensheathing glia can wrap axons as they connect with peripheral organs, associate with dorsal neurons, and encase the neuropil (Fig. 1j).

**Electron microscopic analysis of larval ensheathing glia.** To further study the morphological characteristics of the ensheathing glia, we analyzed serial section transmission electron microscopy (ssTEM) data sets of a first instar larval ventral nerve cord (L1)[43]. We annotated all neuropil-associated glial cells in the abdominal neuromeres A1-A8/9 using CATMAID[44] (Fig. 2). In total, 159 neuropil-associated glial cells were identified that based on their typical morphology could be assigned to one of the three classes of neuropil-associated glial cells[27]: astrocyte-like glial cells, ensheathing glia and ensheathing/wrapping glia (Fig. 2a, b). The morphological characteristics of the different glial subtypes are already evident in first larval instar and become more pronounced in third instar larval brain (Fig. 2c, d). Astrocyte-like glial cells were identified by a prominent process extending into the neuropil proximal to the astrocyte nucleus[29,31]. In the abdominal ventral nerve cord of the L1 volume, 96 astrocyte-like glial cells were identified matching the previously known numbers[27,29].

In total 28 ensheathing glial cells were identified in the abdominal neuromeres 1–7 (Fig. 2a). These glial cells extend thin membrane sheaths along the face of the neuropil (Fig. 2e). The nuclei of the central ensheathing glial cell in each neuromere is usually associated with the dorso-ventral channel[25]. The other ensheathing glia is found at a ventral position at the neuropil cortex interface (see also[27]).

In total 33 cells were annotated as ensheathing/wrapping glial cells, which encase the neuropil and wrap axons in the nerve root. However, this is not very pronounced in the first instar stage, yet (Fig. 2g). With one exception (Fig. 2b), two cells, both located at the dorsal aspect of the neuropil, are associated with each intersegmental nerve root in neuromeres 1–7 (Fig. 2b). The fused A8/A9 nerve has two nerve roots associated with two ensheathing/wrapping glial cells, each (Fig. 2b).

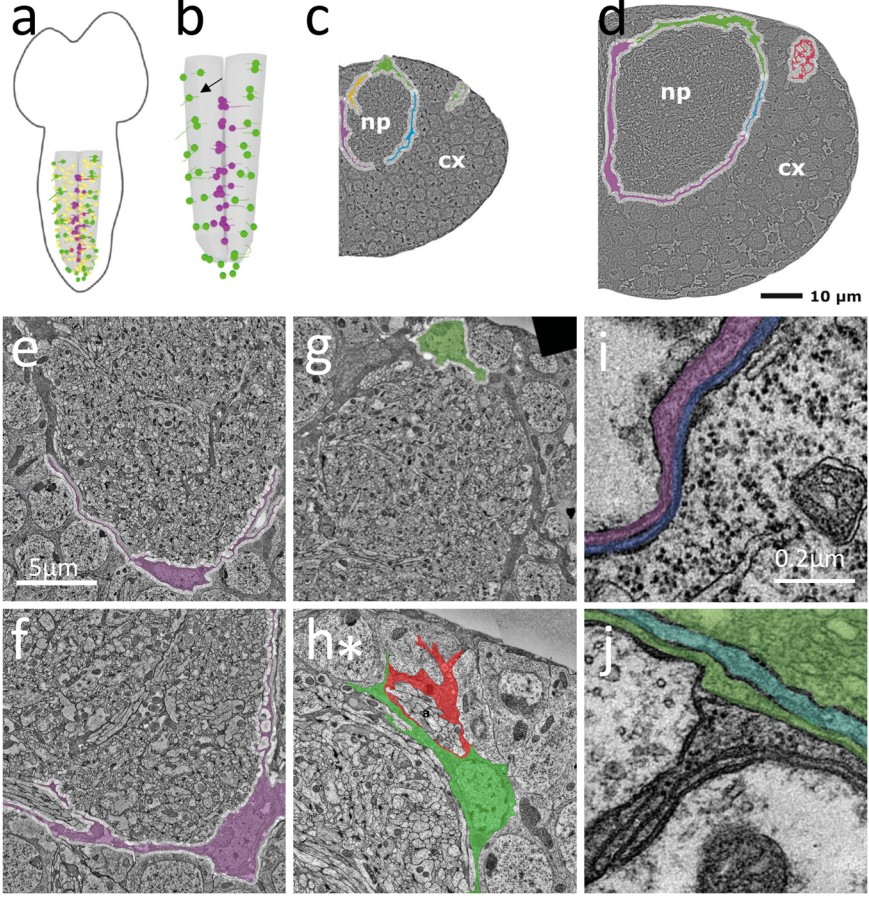

**Fig. 2 Larval development of the ensheathing glia. a, b** Schematic view on a first instar larval central nervous system. All neuropil-associated glial cells were annotated in a serial section TEM volume. Yellow dots indicate the position of the nuclei of astrocyte-like cells, magenta dots indicate the positions of the ensheathing glial cell nuclei, green dots indicate the positions of the ensheathing/wrapping glial nuclei. The arrow points to a segmental position where only one instead of two ensheathing/wrapping glial cell was identified. **c** Representative section through the fourth abdominal hemineuromer of a L1 larva showing the position of three different ensheathing glial cells around the neuropil. **d** Representative section through the fourth abdominal neuromere of a third instar larval ventral nerve cord. Note that ensheathing glia completely cover the neuropil. It was not determined whether the red marked glial processes belong to the blue or green ensheathing/wrapping glial cell. Scale bar (**c, d**): 10 μm, (**e–h**): 5 μm, (**i, j**): 0.2 μm. **e** Ensheathing glia in a first and (**f**) in a third instar larva. **g** Ensheathing/wrapping glial cells in a first and (**h**) in a third instar larva. Note the thin processes that engulf cell bodies of dorsally located neurons (asterisk). **i** On the ventral face of the neuropil an ensheathing glial cell and a cortex glial cell form a two-layered sheath between cortex and neuropil. **j** A highly multilayered glial cell layer is found at the dorsal face of the neuropil.

To determine whether the overall morphology of ensheathing glial cells changes during development, we annotated the ensheathing glia in three abdominal neuromeres (A1-A3) of a third instar larval brain (L3)[45]. Here, too, two ensheathing and two ensheathing/wrapping glial cells are found in each hemi-neuromere (Fig. 2f, h). Note, that dorsally to the neuropil, ensheathing glia send protrusions to the blood–brain barrier and thus adopt a more complex polarized morphology (Fig. 2h). In consequence, dorsal neurons are encased by ensheathing glia processes and not by cortex glia (Fig. 1j, Fig. 2h, see below). The ventrolateral ensheathing glial cells do not encase neuronal cell bodies, but form very flat membrane sheets around the neuropil (Fig. 1g–i, Fig. 2e, f).

The light microscopic analysis suggests that ensheathing glial cells encase the neuropil in third instar larvae. This idea is supported by the ssTEM data sets. In L1 ventral nerve cord ensheathing, ensheathing/wrapping and astrocyte-like glial cells form an almost closed structure around the neuropil (Fig. 2c). In contrast, in the L3 larval ventral nerve cord the neuropil encasement is complete (Fig. 2d). In addition, multiple layers of glial processes are found around the neuropil (Fig. 2i, j). At the ventral face of the neuropil, the glial sheath is often made of cell processes of ensheathing glial and cortex glial cells (Fig. 2i).

**The complexity of ensheathing glia increases during development.** About 112 ensheathing glial cells expressing *83E12-Gal4* can be detected in the third instar larva with only few ensheathing glial cells in the brain lobes (*n* = 5, with 110, 112, 112, 114, and 116 *83E12-Gal4* positive cells, median = 112) (Fig. 3a, m), which corresponds to previously determined numbers[27,35,40,46,47] and matches the annotations made in the TEM volume mentioned above. During the subsequent pupal development, the number of the *83E12-Gal4* positive ensheathing glial cell population increases more than 1,100 ensheathing glial cells in the adult (ventral nerve cord and brain without optic lobes, number determined using Imaris) (Fig. 3c, n).

It is unclear whether all larval ensheathing glia degenerate or whether some survive to participate in the formation of adult ensheathing glia[35,46]. To test whether ensheathing glial cells are competent to divide, we expressed activated FGF-receptor which is known to trigger perineurial glial proliferation[48,49]. Interestingly, a proliferative response upon activated FGF-receptor expression is most prominently observed in thoracic neuromeres (Fig. 3a–f, m, n, Supplementary Fig. S2). We then blocked cell division in the ensheathing glia by RNAi-based silencing of *string*, which encodes the *Drosophila* Cdc25 phosphatase homolog required for progression from G2 to M phase[50]. This expression regime did not severely affect ensheathing glia cell number in the larval CNS (Fig. 3g, h, m, 94 ensheathing glia in *string* knockdown larvae vs. 112 ensheathing glial cells in control animals, *n* = 5 with 83, 83, 110, 99, and 94 *83E12-Gal4* positive cells, median = 94). In contrast, suppression of *string* function resulted in 38% less ensheathing glial cells in the adult nervous system (724 [*n* = 5, 724, 707, 769, 633, and 848 *83E12-Gal4* positive cells, median = 724] instead of the expected 1136 ensheathing glial cells (Fig. 3i, n). Similarly, when we expressed the cell cycle regulator Fizzy related (Fzr) which blocks cell division upon overexpression[51–53] we noted a similar reduction in the number of adult ensheathing glia (Fig. 3j–n). Interestingly, the block of cell proliferation in ensheathing glia mostly affects the adult brain (Fig. 3i, l). To further test whether larval ensheathing glia could divide, we analyzed the ploidy of the ensheathing glia using DAPI staining[54]. In the larval ventral nerve cord, 30 of 30 tested abdominal glial nuclei are polyploid and thus are likely not to divide (Fig. 3o, Supplementary Fig. S3a–d). In

contrast, 10 out of 25 ensheathing glial cells in thoracic neuromeres are diploid and possibly could divide (Fig. 3o, Supplementary Fig. S3a–d). Phospho-histone H3 serves as a specific marker for mitosis. In the larval CNS, we only rarely detected *83E12-Gal4* and phospho-histone H3 positive cells (Supplementary Fig. S3e). Dividing ensheathing glial cells were easily detected in early pupae but not in 42 APF old pupae (Supplementary Fig. S3f–h). In addition, we fed the thymidine analogue 5-ethynyl-2′-deoxyuridine (EdU) which allows to label DNA synthesis during larval development and frequently detected EdU staining in *83E12-Gal4* positive ensheathing glia of larval and young pupal brains (Supplementary Fig. S3i, j). Moreover, when we labelled larval ensheathing glia using a MCFO2 strategy[55] in third instar larvae or early pupae, we found labelled ensheathing glia in the adult (Supplementary Fig. S4a-c). In conclusion, at least some larval *83E12-Gal4* expressing ensheathing glial cells are diploid and can proliferate.

**Ensheathing glial cells are not required for viability.** To test the functional relevance of the ensheathing glia, we performed ablation experiments. Expression of *hid* and *reaper* (*rpr*) using *repo-Gal4* or *83E12-Gal4* resulted in early larval lethality. Since *83E12-Gal4* is also expressed in the midgut, we used a split-Gal4 approach to restrict Gal4 expression to only the ensheathing glial cells of the CNS[56,57]. We generated a construct that allowed the expression of a Gal4 activation domain under the control of the *83E12* enhancer [*83E12-Gal4AD*]. When crossed to flies that carry an element that directs expression of the Gal4 DNA binding domain in all glial cells [*repo-Gal4DBD*], Gal4 activity will be reconstituted only in CNS ensheathing glial cells. Indeed, animals carrying both transgenes [*83E12-Gal4AD, repo-Gal4DBD*] show the same expression domain as the original *83E12-Gal4* driver in the adult as well as in the larval nervous system (Supplementary Fig. S4d–g).

Flies expressing *hid* and *rpr* only in the ensheathing glia [*UAS-rpr; 83E12-Gal4AD; repo-Gal4DBD, UAS-hid*] survive to adulthood. Possibly, in flies with the genotype [*UAS-rpr; 83E12-Gal4AD; repo-Gal4DBD, UAS-hid*] low levels of Gal4 based *hid* and or *rpr* expression might also occur in the related astrocyte-like cells. To further validate the specificity of ensheathing glia ablation, we stained the specimens for Rumpel and Nazgul. Rumpel is a SLC5A transporter strongly expressed by the ensheathing glia and weakly by astrocyte-like glia (Fig. 4). Nazgul is a NADP-retinol dehydrogenase that is specifically expressed by astrocyte-like cells[58]. In control, third instar larval brains Rumpel is found along the entire neuropil with Nazgul expressing astrocytes positioned in a characteristic pattern on the dorsal surface of the neuropil (Fig. 4a–c). Likewise, when we ablated the ensheathing glia in a background of a *UAS-CD8::GFP* transgene [*UAS-rpr; 83E12-Gal4AD; repo-Gal4DBD, UAS-hid*] no ensheathing glia cells can be detected in the CNS. Only few wrapping glial cells in the peripheral nervous system can still be detected (Supplementary Fig. S4h–j).

Astrocytes infiltrate the entire neuropil and form only few processes on the outer surface of the neuropil (Fig. 4c). Upon ablation of the ensheathing glia, only a very faint astrocytic Rumpel signal is detected in the head region (Fig. 4d). In the ventral nerve cord, astrocyte-like cells form additional large cell processes around the neuropil (Fig. 4f). In addition, in adult flies lacking ensheathing glia we noted a 20% increase in the number of Nazgul positive astrocyte-like cells (Fig. 4g–i, *n* = 6 for each genotype, [control: 427, 459, 379, 458, 368 and 433, median = 430; ablation: 507, 520, 469, 478, 516, and 545, median = 512; **$p$ = 0.0022). Adult flies are fertile but lifespan is reduced by about 40% compared to control animals (Fig. 4j, *n* = 200 mated

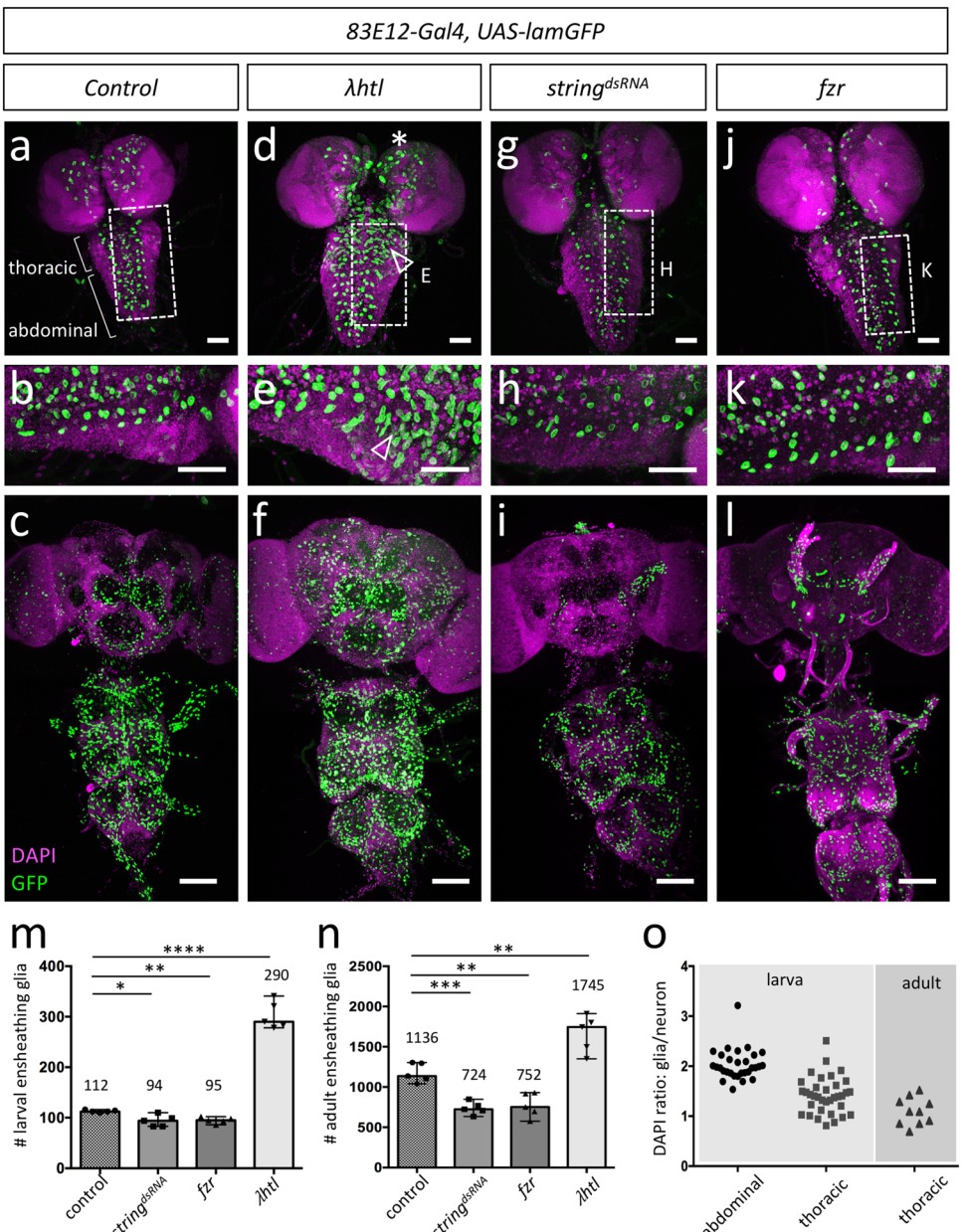

**Fig. 3 Larval thoracic ensheathing glial cells can divide to generate adult brain ensheathing glia.** Representative images are shown. **a** Control third instar larvae with nuclei of ensheathing glia labelled [*83E12-Gal4, UAS-Lam::GFP*]. Nuclei are stained using DAPI. The boxed area is shown in high magnification in (**b**). **c** Adult control brain. The number of ensheathing glial nuclei is quantified in (**m, n**, *n* = 5 brains for all genotypes). **d–f** Expression of an activated FGF-receptor leads to an increase in the number of larval ensheathing glia in larval thoracic neuromeres (arrowheads). The white boxed area is shown in higher magnification (**e**). **g–i** Upon expression of *string* dsRNA in all ensheathing glia the number of larval ensheathing glial cells is slightly reduced. For quantification see (**m**). The boxed area in the larval CNS (**g**) is shown in high magnification in (**h**). **i** Expression of *string* dsRNA in all ensheathing glia reduces the number of ensheathing glial cells in the adult CNS. **j–l** A similar reduction in the number of ensheathing glial cells is observed following expression of *fzr*. The boxed area in (**j**) is shown in high magnification in (**k**). Scale bars larval CNS (**a, b, d, e, g, h, j, k**) are 50 μm, scale bars for adult CNS (**c, f, i, l**) are 100 μm. Quantification of the ensheathing glial cell number in five larval brains (**m**) and in five adult brains (**n**) using Imaris (unpaired t-test, two-tailed). The standard deviation is indicated. The optic lobes and the tract ensheathing glial cells were excluded in the quantification. For larva: control – *string^{dsRNA}*: *$p$ = 0.0192; control – *fzr*: **$p$ = 0.0013; control - *λhtl*: ****$p$ =< 0.0001; for adult: control – *string^{dsRNA}*: ***$p$ = 0.0002; control - *fzr*: **$p$ = 0.0018; control - *λhtl*: **$p$ = 0.0057. (O) Quantification of DAPI intensity in 30 larval abdominal, 30 larval thoracic, and 10 adult thoracic ensheathing glial and neuronal nuclei. Source data are provided as a Source Data file.

females, $p$ = 4,67365E-83). In conclusion, these data show that ensheathing glial cells are not essential for viability. Upon ablation of ensheathing glia, astrocyte-like cells form additional processes encasing the neuropil. This compensatory growth of astrocyte-like cells suggests that barrier establishment is a key function of the ensheathing glia.

**Ensheathing glial cells contribute to barrier formation around the neuropil.** The third instar larval CNS is covered by subperineurial glial cells that do not allow penetration of labelled 10 kDa dextran. In wild type, subperineurial glial cells block paracellular diffusion across the blood–brain barrier by forming septate junctions[59]. However, in larvae lacking septate junctions,

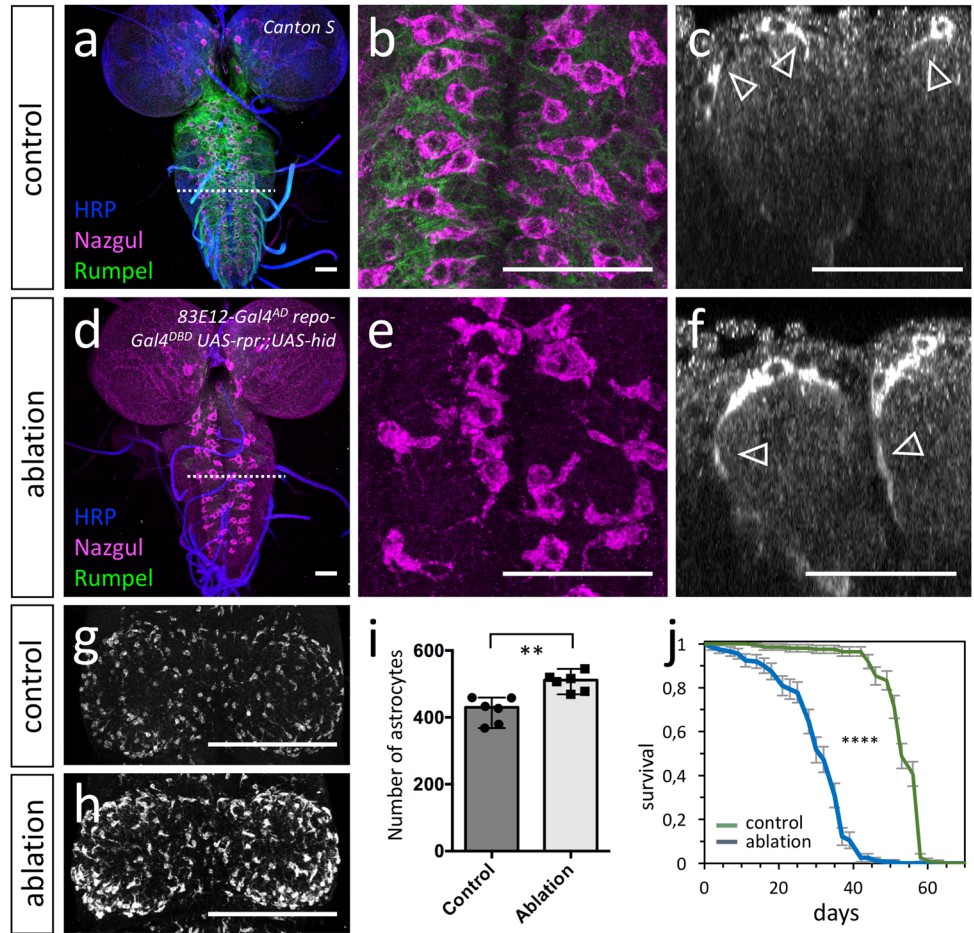

**Fig. 4 Ablation of ensheathing glia causes compensatory growth and increased proliferation of astrocyte-like cells.** Representative images are shown. **a**–**f** Third instar larval brains stained for Rumpel (green, expressed by ensheathing glia), Nazgul (magenta, expressed by astrocyte-like cells) and HRP (blue, neuronal membrane marker). **a** Maximum projection of a confocal stack of a control larval CNS. The dashed line indicates the position of the orthogonal view shown in (**c**). **b** Dorsal view of a control ventral nerve cord. Note the regular positioning of the astrocyte-like cells (magenta). **c** Orthogonal section of astrocyte-like cells which are labelled using anti-Nazgul staining (white, arrowheads point to short protrusions). **d, e** Upon ablation of the ensheathing glia [*UAS-rpr; 83E12-Gal4^{AD}; repo-Gal4^{DBD}, UAS-hid*], Rumpel expression is greatly reduced. The dashed line indicates the position of the orthogonal view shown in (**f**). **e** Dorsal view of a larval ventral nerve cord following ensheathing glia ablation. The position and the morphology of the astrocyte-like cells appears disorganized. **f** Orthogonal section. Note, that astrocyte-like glial cells develop long cell processes that appear to encase the entire neuropil in the absence of ensheathing glia (arrowheads). **g** Dorsal view on a thoracic neuromere of an adult control fly. **h** Thoracic neuromere of an adult fly lacking ensheathing glial cells. The number of astrocyte-like cells increases. **i** Quantification of the number of astrocyte-like cells in the ventral nerve cord of adult flies ($n = 6$ brains, $**p = 0.0018$, unpaired t-test, two-tailed, standard deviation is indicated). **j** Upon ablation of the ensheathing glia, longevity is reduced ($n = 200$ mated females, $****p = 4.67365E-83$, Log-rank test, two tailed, standard deviation is indicated). Scale bars are: **a**–**f** 50 μm, **g**, **h** 100 μm. Source data are provided as a Source Data file.

extensive interdigitations between neighboring subperineurial cells can also provide a barrier function[60].

Extensive electron microscopic analyses failed to demonstrate septate junction like structures between different ensheathing glial cells. However, glial processes overlap extensively at the neuropil cortex interface (Fig. 2i, j), which might increase the length of the diffusion path at the boundary between CNS cortex and neuropil. To directly test whether ensheathing glial cells indeed provide a barrier function, we performed dextran diffusion assays in control brains and those lacking ensheathing glia. For this, we carefully dissected third instar larval brains including anterior cuticular structures and placed them on a coverslip. Upon injection of fluorescently labelled 10 kDa dextran directly into the neuropil of one brain lobe employing capillary normally used for DNA injections, we determined diffusion of the labelled dextran within the neuropil through the large brain commissure into the contralateral hemisphere (Fig. 5a). Diffusion of dextran was

monitored under the confocal microscope and quantified for 10 minutes (Fig. 5b, d, movies S1,2). The ratio of the fluorescence measured at two identical sized ROIs at the neuropil and the cortex contralateral to the injection site was plotted against time (Fig. 5c). In control larvae, a fluorescence is mostly confined to the neuropil area and only little dye reaches the cortex area. When we performed the injection experiments in larvae lacking ensheathing glia, we noted a significantly faster increase in fluorescence signal in the cortex area relative to the neuropil (Fig. 5c). Thus, we conclude that the ensheathing glia, although they lack specialized occluding junctions, contribute to a barrier function that possibly involves the extensive overlap noted for ensheathing glia cell processes (Fig. 2).

**Ensheathing glial cells are polarized.** Tissue barriers are generally formed by polarized cells which are characterized by a

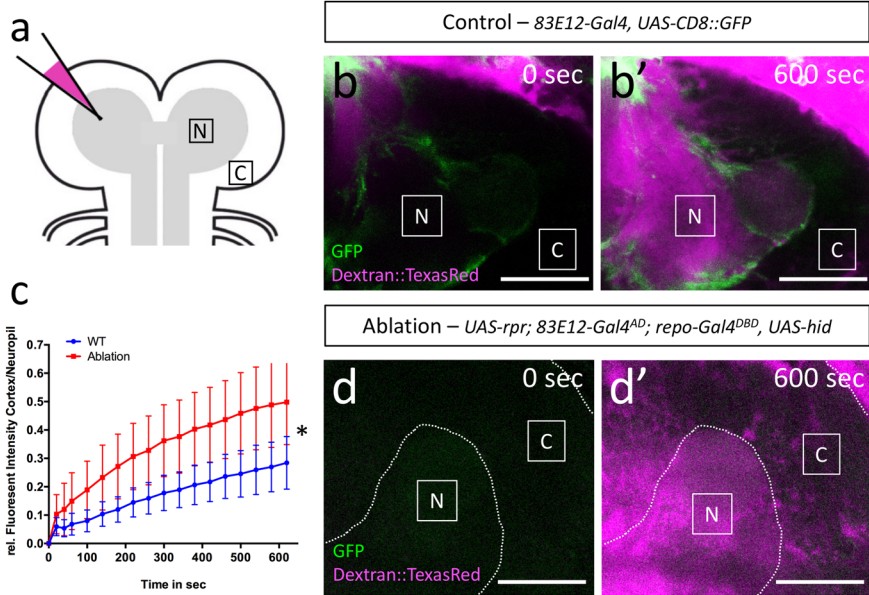

**Fig. 5 Ensheathing glial cells form an internal barrier around the neuropil. a** Schematic depiction of the dye injection experiments. Fluorescently labelled 10 kDa dextran was injected into the left hemisphere of a wandering third instar larva. Diffusion of the dye was monitored at two positions. In the neuropil (N) and in the cortex (C) harboring all neuronal cell bodies. **b, b'** Stills of a representative movie of a larval brain lobe with the genotype [$n = 5$; *83E12-Gal4, UAS-CD8::GFP*] following dye injection, time is indicated. **c** Quantification of dye diffusion rate in larvae of the genotype [*83E12-Gal4, UAS-CD8::GFP*] (blue line) and [*UAS-rpr; 83E12-Gal4AD, repo-Gal4DBD, UAS-hid*] (red line) ($n = 5$, standard deviation is shown, $*p = 0.0181$; unpaired *t*-test, two-tailed). **d, d'** Stills of a representative movie of a larval brain lobe with the genotype [$n = 5$, *UAS-rpr; 83E12-Gal4AD; repo-Gal4DBD, UAS-hid*] following dye injection. Scale bars are 50 μm. Source data are provided as a Source Data file.

differential distribution of plasma membrane lipids. The apical membrane is generally rich in phosphatidylinositol 4,5-bisphosphate (PIP₂), whereas the basolateral cell membrane contains more phosphatidylinositol 3,4,5-triphosphate (PIP₃)[61,62]. To detect a possible differential localization of these phospholipids, we employed different lipid sensors based on distinct PH-domains. PH-PLCδ-mCherry is targeted by PIP₂[63] whereas PH-AKT-GFP preferentially binds PIP₃[64]. We focused our analysis on the ensheathing glial cells covering the dorsal aspect of the neuropil, since the ventrolateral ensheathing glia form very thin, less than 200 nm thick, sheets around the neuropil that precludes a further analysis by confocal microscopy.

We co-expressed both sensors in the ensheathing glia using *83E12-Gal4* and quantified the distribution of both sensors by determining the number of green and red fluorescent pixels in *83E12-Gal4* positive cells in orthogonal sections comparing the cell domain close to the neuropil with the one close to the blood–brain barrier (Fig. 6a–g). Here, we noted an increased PH-PLCδ-mCherry localization at the direct interface of the neuropil and the ensheathing glia (Fig. 6b–e). For quantification, we measured the mean fluorescence intensity of PH-PLCδ-mCherry and PH-AKT-GFP (30 dorsal neuropil areas of 10 larvae). This demonstrated an almost twofold enrichment of PIP₂ at the plasma membrane domain facing the neuropil. The PIP₃ detecting PH-AKT-GFP shows a complimentary distribution with an enrichment in basolateral plasma membrane domains (Fig. 6f, g). A similar polarization of the ensheathing glial cells can be detected in adult stages (Fig. 6f; Supplementary Fig. S5, 30 neuropil areas in 10 adult brains).

Polarized epithelial cells are characterized by an enrichment of the Na⁺/K⁺-ATPase Nervana 2 (Nrv2) at the basolateral plasma membrane[65]. To study the localization of Nrv2, we employed a gene trap insertion[66]. As found for PIP₃, Nrv2::GFP localizes predominantly at the basal side of the ensheathing glia (Fig. 6h–k), which corresponds to the localization of Nrv2 in

epithelial cells. Therefore, we conclude that dorsal ensheathing glial cells are polarized cells throughout development, facing their apical-like domain towards the neuropil (Fig. 6g). Although all polarity markers are also expressed by the ventrolateral ensheathing glia, we cannot resolve their polarity due to the very flat shape of these membrane sheets which does not allow the separation of apical and basolateral membrane domains by optical methods.

**Integrin α and extracellular matrix components are expressed by adult ensheathing glia.** In epithelial cells, apical-basal polarity is also reflected by a basally located integrin receptor which bind the basally located extracellular matrix (ECM) proteins. *inflated* encodes an alpha-subunit of the integrin receptor and *inflated* mRNA is expressed by adult perineurial and ensheathing glial cells[11,67] (Supplementary Fig. S6). Endogenously YFP-tagged Integrin α (if^CPTI-004152 [68]) localizes at the blood–brain barrier and around the neuropil in larval brains (Fig. 7a, b, e). In the adult brain, Integrin^YFP α localization around the neuropil appears more pronounced compared to abdominal neuromeres (Fig. 7c, d). Integrin receptors bind ECM components, suggesting that these proteins are also expressed within the nervous system. We thus tested expression of all genes annotated to encode proteins involved in extracellular matrix formation (FlyBase) using published single cell sequencing data of adult brains[11] (Supplementary Fig. S6). Eight of these genes appear expressed by the ensheathing glia and GFP-protein trap insertion lines were available for *trol* (Perlecan), *viking* (Collagen IV), and *dally* (Glypican).

The heparan sulfate proteoglycan Dally is most strongly detected close to the cells of the larval blood–brain barrier (Fig. 7f). Within the larval CNS, Dally is found in the cortex and is enriched around the larval neuropil (Fig. 7f, g). In adults, Dally localization at the blood–brain barrier ceases but is still found in

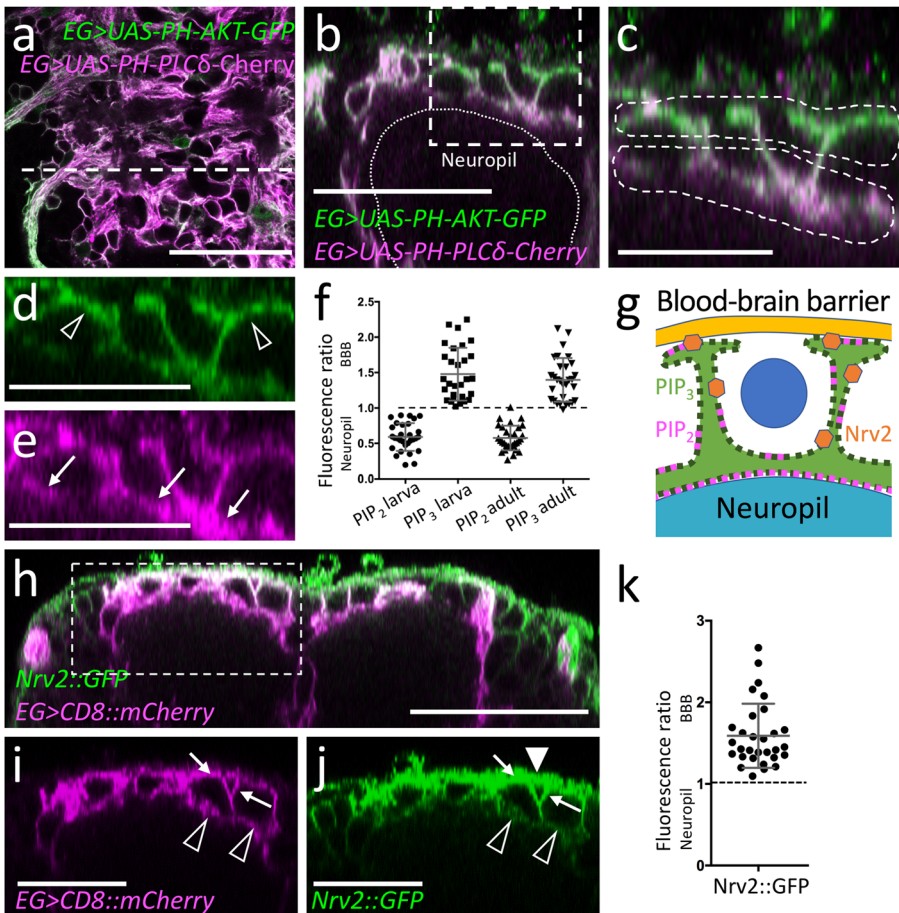

**Fig. 6 Ensheathing glial cells show polarized plasma membrane domains.** Representative images are shown. **a** Projection of a confocal stack of a third instar larval ventral nerve cord coexpressing *PH-AKT-GFP* and *PH-PLCδ-mCherry* in ensheathing glial cells (EG) [*83E12-Gal4, UAS-PH-AKT-GFP, UAS-PH-PLCδ-mCherry*]. The dashed white line indicates the position of the orthogonal section shown in (**b**). **b** Orthogonal section showing the dorsal aspect of the neuropil. The boxed area is shown in higher magnification in (**c**–**e**). The dashed areas were subsequently used for quantification of GFP and mCherry localization. **f** Quantification of GFP/mCherry distribution in larval and adult ensheathing glia. The ratio of red (PLCδ-mCherry shown in magenta) and green (PH-AKT-GFP) fluorescent pixels of the areas indicated in (**c**) is plotted. The mean and standard deviation is shown. **g** Schematic view of a dorsal ensheathing glial cell. The neuropil facing domain is characterized by a high PIP$_2$ content, whereas PIP$_3$ is concentrated on the baso-lateral domain, where Nrv2 is predominantly localized, too. The blue dot indicates the position of a neuron. **h** Coexpression of *Nrv2::GFP* and *83E12-Gal4, UAS-mCherry* in the ventral nerve cord of third instar larva. The boxed area is shown in higher magnification in (**i**, **j**). Note the preferential localization of Nrv2 at the basolateral cell domain of the ensheathing glia (arrows). Only little Nrv2 is found at the apical domain (open arrow head). Additional expression of Nrv2 is seen in the blood–brain barrier (filled arrowhead). **k** Quantification of polarized Nrv2::GFP localization. The mean and standard deviation is shown. Scale bars in **a**, **b**, **h**: 50 µm; in **b**–**e**, **i**, **j**: 25 µm. Source data are provided as a Source Data file.

the CNS cortex. A prominent enrichment of Dally is seen around the neuropil (Fig. 7h–j). In contrast to Dally, we could not detect Trol and Viking around the larval neuropil and only detected strong signals at the blood–brain barrier (Fig. 7k, l, p, q). However, in the adult nervous system both Trol and Viking are detected basally of the ensheathing glia facing the cortex glia with an even distribution in the different parts of the CNS (Fig. 7m–o, r–t). In conclusion, the ensheathing glia express ECM proteins which is characteristic for polarized cell types.

**Polar distribution of ß$_{Heavy}$-Spectrin in larval ensheathing glia.** Polarization of cells is also evident in the cytoskeleton underlying the plasma membrane. Whereas α-Spectrin is found below the entire plasma membrane, ß-Spectrin decorates the basolateral plasma membrane and ß$_{Heavy}$-Spectrin (ß$_H$-Spectrin) is located at the apical domain of the cell[69]. ß-Spectrin is found in all neural cells in the ventral nerve cord[70] and no specific localization can be resolved in the ensheathing glia due to its strong overall expression.

ß$_H$-Spectrin is encoded by the *karst* gene. One available MiMIC insertion allows to target two of the seven isoforms (Fig. 8a). In third instar larval brains, GFP-ß$_H$-Spectrin localizes at the blood–brain barrier (Fig. 8c, d, f, g), and close to the neuropil (Fig. 8c, d, f), indicating expression by the ensheathing glia. Here, ß$_H$-Spectrin is enriched close to the apical-like, PIP$_2$ containing membrane of the ensheathing glia facing towards the neuropil (Fig. 8c, d). To further validate the expression of Karst in the ensheathing glia, we silenced *karst* expression by expressing double stranded RNA. When *karst* expression is silenced in all glial cells using *repo-Gal4* only expression in trachea as well as a weak background signal at the neural lamella around the CNS is detected (Supplementary Fig. S7a, b). Silencing of *karst* expression in the blood–brain barrier using *moody-Gal4* did not affect localization of GFP around the neuropil (Supplementary Fig. S7c, d). When we silenced *karst* specifically in the ensheathing glia using *83E12-Gal4* protein localization in the blood–brain barrier is unaffected but no ß$_H$-Spectrin protein can be detected around the neuropil (Fig. 8g).

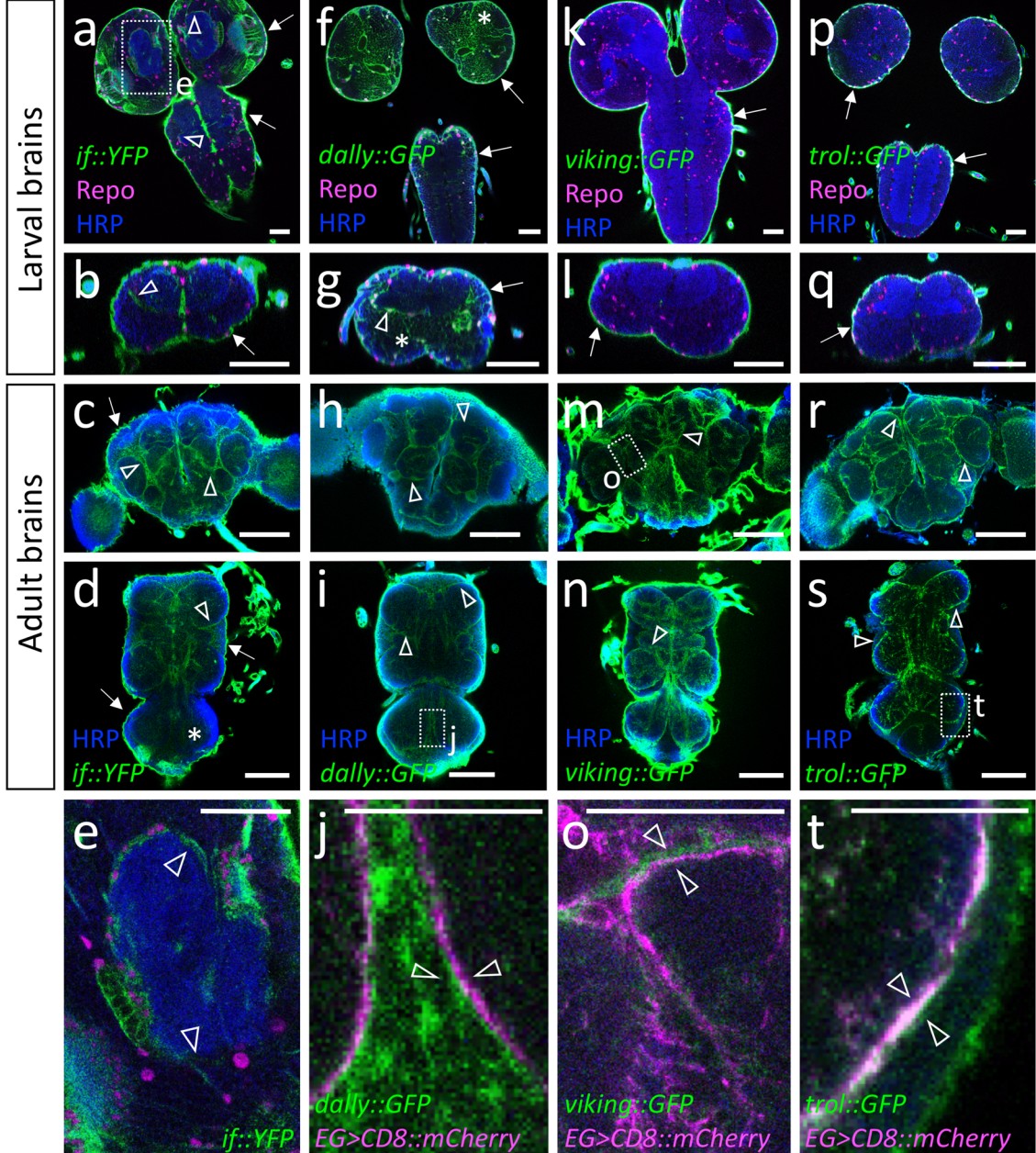

**Fig. 7 The ensheathing glial cells are flanked by extracellular matrix.** Representative images are shown. Expression of Integrin α and different extracellular matrix components as detected by using gene trap insertion lines. **a–e** Endogenously YFP-tagged Integrin α encoded by the *inflated* gene. **f–j** Endogenously GFP-tagged heparan sulfate proteoglycan Dally protein encoded by *dally::GFP*. **k–o** Endogenously tagged Collagen IV protein encoded by *viking::GFP*. **p–t** Endogenously tagged Perlecan encoded by *trol::GFP*. The different developmental stages are indicated. **a, i, m, s** The boxed areas are shown in the bottom row. Glial cells are in magenta (anti-Repo staining), neuronal membranes are in blue (anti-HRP staining) and the GFP-tagged proteins are in green (anti-GFP staining). **a, b, e** During late larval stages, prominent Inflated expression is seen at the blood–brain barrier (arrows) and weak expression is seen around the neuropil at the position of the ensheathing glia (arrowhead). **c, d** In adults, Integrin α is still found at the blood–brain barrier (arrows) and becomes more prominent around the neuropil (arrowheads). Note that strongest Inflated expression is detected in larval brains. No Inflated expression can be detected in the abdominal neuromeres (asterisk). **f, g** During the larval stage Dally is expressed at the blood–brain barrier (arrows) and in the CNS cortex (asterisks). A slightly stronger localization of Dally can be detected at the position of basal ensheathing glia processes (arrowhead). **h–j** In adults, Dally is enriched at the position of the ensheathing glia (arrowheads). **k, l** In the larval CNS Collagen IV is found at blood–brain barrier (arrows) but not within the nervous system. **m–o** Collagen IV is detected at the neuropil-cortex interface in adult stages (arrowheads in **m, n**, boxed area is shown in (**o**)). **p, q** Trol is detected at the larval blood–brain barrier (arrows) but not around the neuropil. **r–t** In adults, prominent Trol localization is seen at the position of the ensheathing glia (arrowheads). The boxed area in (**s**) is shown at higher magnification in (**t**). Scale bars are: larval CNS 50 µm, adult CNS 100 µm except for **e**: 25 µm; **j**, **o**, **t**: 50 µm.

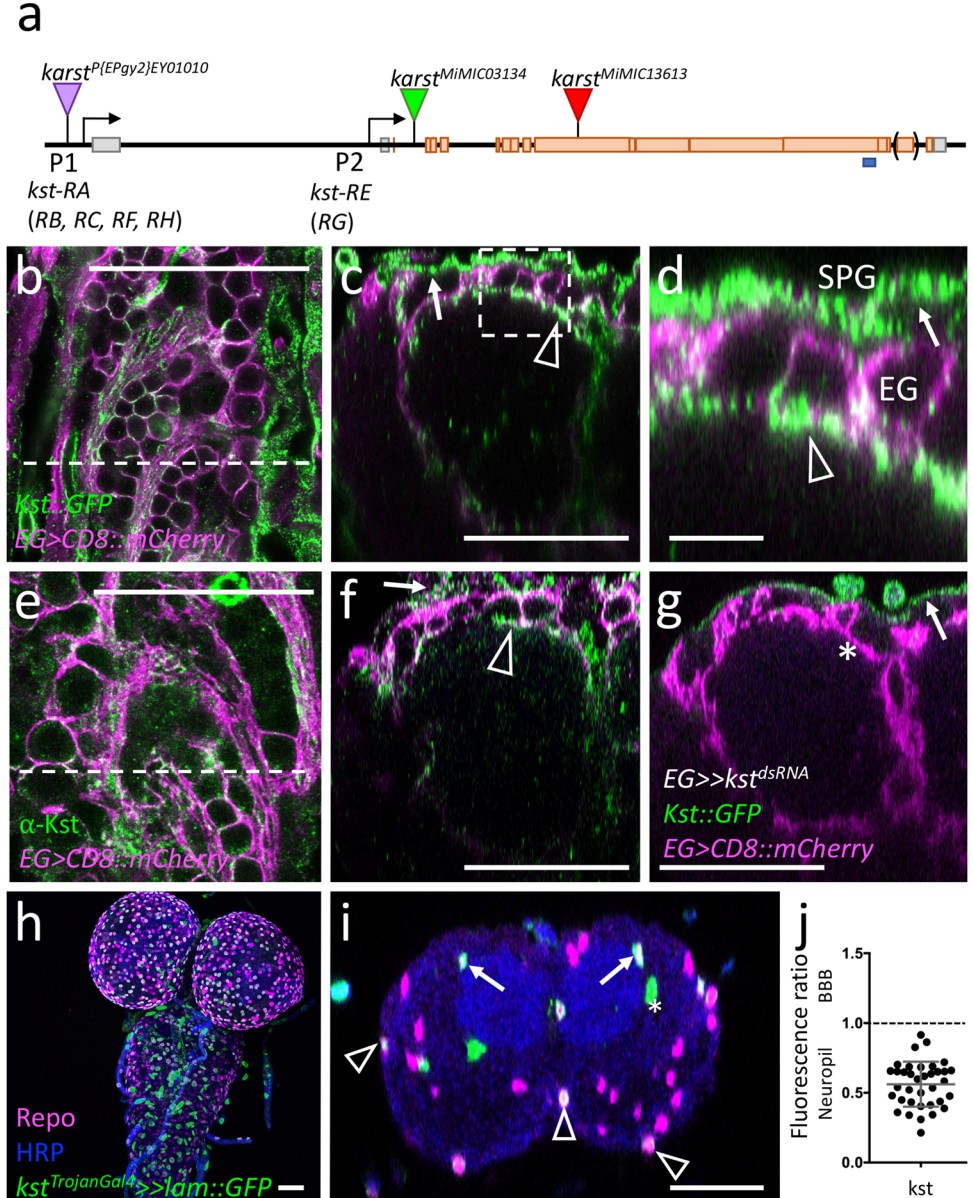

**Fig. 8 ß_H-Spectrin shows a polar distribution in ensheathing glia.** Representative images are shown. **a** Schematic view of the *karst* locus. Transcription is from left to right. Seven different *karst* mRNAs are generated from two promoters (P1, P2) as indicated (*kst-RA - RH*). The ß_H-Spectrin proteins PE and PG differ in a C-terminal exon indicated by brackets. The position of two MiMIC insertions and the EP insertion used for gain of function experiments is indicated, the blues line denotes the position of the peptide used for immunization. **b** Confocal view of the surface of a third larval instar brain of a *kst^MiMIC03134::GFP* animal that expresses mCherry in all ensheathing glia [*83E12-Gal4, UAS-CD8::mCherry*]. The dashed line indicates the position of the orthogonal view shown in (**c, d**). Expression at the blood–brain barrier forming subperineurial glia (arrows) and at the apical domain of the ensheathing glia (arrowheads) can be detected. For quantification see (**k**). **e** Similar view as shown in (**b**) of a control larva stained for Kst expression using a polyclonal antiserum. The dashed line indicates the position of the orthogonal view shown in (**f**). **f** Localization of Kst protein is weakly seen in the blood–brain barrier (arrow) and the ensheathing glia (arrowhead). **g** Orthogonal view of a *kst^MiMIC03134::GFP* animal expressing *kst^dsRNA* in the ensheathing glia (EG) [*83E12-Gal4, UAS-kst^dsRNA*]. Expression of Karst in ensheathing glia cannot be detected anymore (asterisk). Expression in the blood–brain barrier is unaffected (arrow). **h, i** The Trojan Gal4 element in *kst^MiMIC03134* directs Gal4 expression in the *kst P2* pattern. Lamin::GFP localizes to ensheathing glial nuclei (arrows), some cells of the blood–brain barrier (arrowheads) and some cells in the position of tracheal cells (asterisks). **j** Quantification of Karst localization in different membrane domains of ensheathing glial cells, quantification as described in Fig. 6. Scale bars are 50 µm except **d**: 10 µm. Source data are provided as a Source Data file.

A similar ß_H-Spectrin localization was detected using antisera directed against a C-terminal peptide present in all predicted ß_H-Spectrin isoforms (Fig. 8a, e, f, for specificity of the generated antiserum see: Supplementary Fig. S7e, f). Furthermore, we generated a Trojan-Gal4 insertion using the same MiMIC insertion that was used to generate the *Karst^GFP* protein trap

(Fig. 8a). The *kst^Trojan-Gal4* allele is expected to report only the expression pattern of the proximal promoter which activates expression of two out of the seven *karst* splice variants (FlyBase) (Fig. 8a). Indeed, when driving a nuclear GFP reporter (*UAS-Lam::GFP*), expression in the ensheathing and blood–brain barrier glial cells is apparent (Fig. 8h, i). In addition, cells in

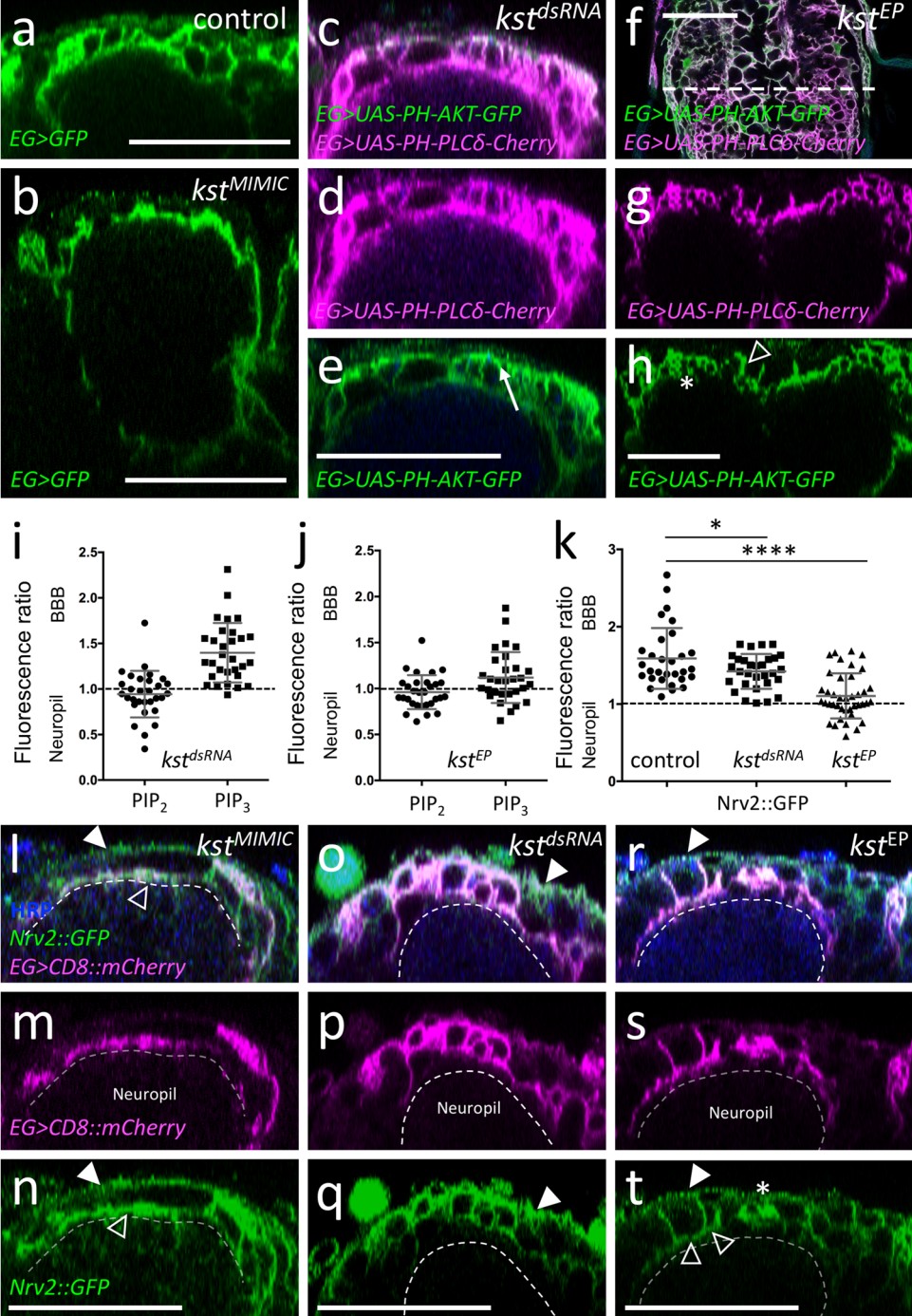

the position of tracheal cells express the *kst^(Trojan-Gal4)* allele (Fig. 8i). During pupal stages, expression of Karst ceases between 72 and 96 hours after puparium formation (APF) (Supplementary Fig. S8). Adult ensheathing glial cells lack detectable expression of the *kst^(MiMIC::GFP)* insertion and show no reactivity using the anti-Karst antisera.

the position of tracheal cells express the *kst^(Trojan-Gal4)* allele (Fig. 8i). During pupal stages, expression of Karst ceases between 72 and 96 hours after puparium formation (APF) (Supplementary Fig. S8). Adult ensheathing glial cells lack detectable expression of the *kst^(MiMIC::GFP)* insertion and show no reactivity using the anti-Karst antisera.

**ß_H-Spectrin is required for ensheathing glial polarity.** We next tested whether ß_H-Spectrin is required for ensheathing glial cell polarity and determined the overall morphology of the ensheathing glia in the *karst* loss of function *karst^(MiMIC13613)* larvae using *83E12-Gal4* driving *CD8::GFP*. In *karst* mutants, larval ensheathing glial cells are still present but show an abnormal collapsed morphology. The extension of ensheathing glia cell protrusions around dorsally located neuronal cell bodies is less evident (Fig. 9a, b). When we silenced *karst* expression using RNAi, we noted a weaker phenotype and dorsal protrusions frequently formed (Fig. 9c). This allowed an analysis of the distribution of PIP_2 and PIP_3 (Fig. 9c–e). Whereas in ensheathing glial cells of control ventral nerve cords, PIP_2 (as detected by PH-PLCδ-mCherry) is found predominantly on the domain facing the neuropil, an almost even distribution is noted upon *kst* knockdown in larvae (compare Figs. 6f, 9i). Interestingly, the polar distribution of PIP_3 does not appear to be affected by *kst* knockdown (Fig. 9e, i). To further test whether ß_H-Spectrin is needed for ensheathing glial polarization, we performed over-expression experiments. To direct expression of wild type Karst

**Fig. 9 The ß$_H$-Spectrin cytoskeleton is required for glial polarity.** Representative images are shown. **a** Orthogonal view of the dorsal aspect of a third instar control ventral nerve cord stained for ensheathing glial (EG) morphology [83E12-Gal4, UAS-CD8::GFP]. **b** Third instar larval brain of a zygotic *karst* null mutant larvae with the genotype [kst$^{MiMIC13613}$ / kst$^{MiMIC13613}$, 83E12-Gal4, UAS-CD8::GFP]. The absence of all ß$_H$-Spectrin protein affects the morphology of the dorsally located ensheathing glial cells. **c–e** Third instar larval ventral nerve cord with reduced *kst* expression in ensheathing glia [83E12-Gal4, UAS-kst$^{dsRNA}$, UAS-PH-AKT-GFP, UAS-PH-PLCδ-Cherry]. **d** PH-PLCδ-Cherry binds to PIP$_2$ and shows an even distribution between apical and basolateral plasma membrane domains. For quantification see (**i**). Quantification was performed as described in Fig. 6. **e** PH-AKT-GFP binds to PIP$_3$ and is distributed in a polarized manner. For quantification see (**i**). **f–h** Third instar larval ventral nerve cord with increased *kst* expression in ensheathing glia [83E12-Gal4, kst$^{P\{EPgy2\}EY01010}$, UAS-PH-AKT-GFP, UAS-PH-PLCδ-Cherry]. Note, the variable phenotype noted upon *karst* overexpression. Ensheathing glia with almost normal morphology (open arrowhead) is next to hyperconvoluted ensheathing glia (asterisk). Quantification of PIP$_2$ and PIP$_3$ localization is shown in (**j**). **i, j** Quantification was performed as described in Fig. 6. The mean and the standard deviation is shown. **k** Quantification of the distribution of Nrv2::GFP using an unpaired t-test, two-tailed. *$p_{control-kstdsRNA}$ = 0.0474; ****$p_{control-kstEP}$ < 0.001. The mean and the standard deviation is shown. **l–n** In *karst* mutant larvae, Nrv2 is found in ensheathing glia cell processes (open arrowhead) that face the neuropil. The dashed line indicates the neuropil ensheathing glia boundary. **o–q** Upon ensheathing glial specific *kst* knockdown, Nrv2 localizes to the basolateral cell domains, for quantification see (**k**). Filled arrowhead indicates the blood–brain barrier. **r–t** Upon overexpression of Karst, polarized localization of Nrv2 is less apparent in ensheathing glial cell processes (normal morphology: open arrowheads, hyperconvoluted morphology: asterisk), for quantification see (**k**). Scale bars are 50 µm. Source data are provided as a Source Data file.

protein in ensheathing glia, we utilized the EP-element insertion P{EPgy2}EY01010. Overexpression of *karst* results in abnormally convoluted and partially collapsed ensheathing glial cell morphology characterized by an even distribution of both PIP$_2$ and PIP$_3$ (Fig. 9f–h, j).

We next assayed Nrv2 localization in *karst* null mutants as well as upon 83E12-Gal4 directed *kst* knockdown. In larvae lacking *karst* expression ensheathing glial cells show a collapsed morphology and polarized localization of Nrv2 cannot be resolved (Fig. 9l–n). Upon *kst* knockdown Nrv2 correctly localizes to basolateral protrusions (Fig. 9k, o–q). In *kst* gain of function larvae, the polarized Nrv2 distribution is lost (Fig. 9k, r–t), supporting the notion that the spectrin cytoskeleton might affect proper positioning of Nrv2[65,71–73].

**Polarized ensheathing glia is required for locomotor behavior.** The above data show that larval ensheathing glia are polarized, ECM abutting cells that separate the neuropil from the CNS cortex and are required for longevity of the adult fly. To test whether the ensheathing glial cells are required for normal locomotor control during larval stages, we compared locomotion of control animals with those lacking ensheathing glia or with those with reduced ß$_H$-Spectrin or Nrv2 expression using FIM imaging[74,75].

Control animals move on long paths interrupted by short reorientation phases that are characterized by increased body bending (Fig. 10a). *kst* knockdown specifically in ensheathing glial cells [83E12-Gal4$^{AD}$; repo-Gal4$^{DBD}$, UAS-kst$^{dsRNA}$] causes a reduction in the peristalsis efficiency during go-phases (Fig. 10a, b, f, g). Likewise, crawling velocity is reduced significantly (Fig. 10g). This suggests that the specific lack of ß$_H$-Spectrin in ensheathing glia causes a strong locomotor phenotype. The observed locomotor phenotypes might be due to an ectopic neuronal activity of the split-Gal4 driver in addition to its activity in the ensheathing glia. Assuming that the *repo* promoter is indeed glial cell specific[76], neuronal activity is not expected in the split Gal4 approach. Moreover, we have no evidence that *karst* is expressed by neurons. Furthermore, we analyzed *kst* mutant larvae to validate the RNAi-induced phenotypes. The split Gal4 driver might have ectopic activity in other glial cells, but given the specific *karst* knockdown only in ensheathing glia we consider this unlikely (Fig. 8g). The Trojan-Gal4 insertion in the *karst$^{MiMIC03134}$* insertion is expected to affect only isoforms Karst-PE and Karst-PG (Fig. 8a). As control, we used an insertion of the Gal4 element in the opposite, unproductive orientation. Similar as detected for the *kst* knockdown, we noted a decreased peristalsis efficiency and a reduced crawling velocity (Fig. 10c, i, j). Larvae completely lacking zygotic

*karst* expression [karst$^{MiMIC13613}$/Df(3 L)ED2083] show a comparable larval locomotion phenotype (Fig. 10d, i, j). This larval locomotion phenotype is similar to the one observed following ablation of the ensheathing glial cells using the genotype [UAS-rpr; 83E12-Gal4$^{AD}$; repo-Gal4$^{DBD}$, UAS-hid] (Fig. 10e). Thus, we conclude that polarized ensheathing glia that connect the blood-brain barrier with the dorsal neuropil are required for normal locomotor behavior. In order to perform vectorial transport, the Na$^+$/K$^+$ ATPase must act in a polarized fashion. We thus also compared larval locomotion of animals with reduced *nrv2* expression to control animals. Interestingly, larvae with ensheathing glial cells lacking *nrv2* expression behave opposite to larvae that lack ß$_H$-spectrin showing an increased peristalsis efficiency as well as an increased crawling velocity (Supplementary Fig. S9). The analysis of the role of polarized Nrv2 distribution for ensheathing glia physiology will thus be an interesting topic for future research.

**Discussion**
Here, we present a comprehensive analysis of the ensheathing glia found in the CNS of *Drosophila*. We developed a split-Gal4 tool to specifically manipulate the ensheathing glia and show that ensheathing glial cells are highly polarized cells with an apical-like cell domain facing towards the neuropil. Polarized membrane lipids as well as several polar localized proteins were identified. The apical sub-membranous cytoskeleton is organized by ß$_H$-Spectrin encoded by *karst*. Ensheathing glia cells contribute to the formation of an internal diffusion barrier as determined by dye penetration experiments in ensheathing glia ablated animals. Flies that lack ensheathing glia have increased numbers of astrocytes, they are viable but show reduced longevity. Moreover, we show that ensheathing glial cells lacking Karst have an altered cell polarity and *kst* mutants as well as cell type specific knockdown larvae show abnormal locomotor behavior.

To study ensheathing glial cell biology, we utilized the 83E12-Gal4 driver which, as described[41], appears specific for ensheathing glia. Some of the larval ensheathing glia are diploid and block of cell proliferation reduces their number by 38%, suggesting that proliferation of these cells contributes to the increase of ensheathing glial complexity. These finding support the notion that some larval ensheathing glia can dedifferentiate, proliferate and redifferentiate to form the more complex adult ensheathing glia[46]. Ablation of ensheathing glia results in an increased number of adult astrocyte-like cells which suggests that the close interdependence of these two cell types identified in embryonic stages[27,77] persists to adulthood. Similarly, a plastic interaction

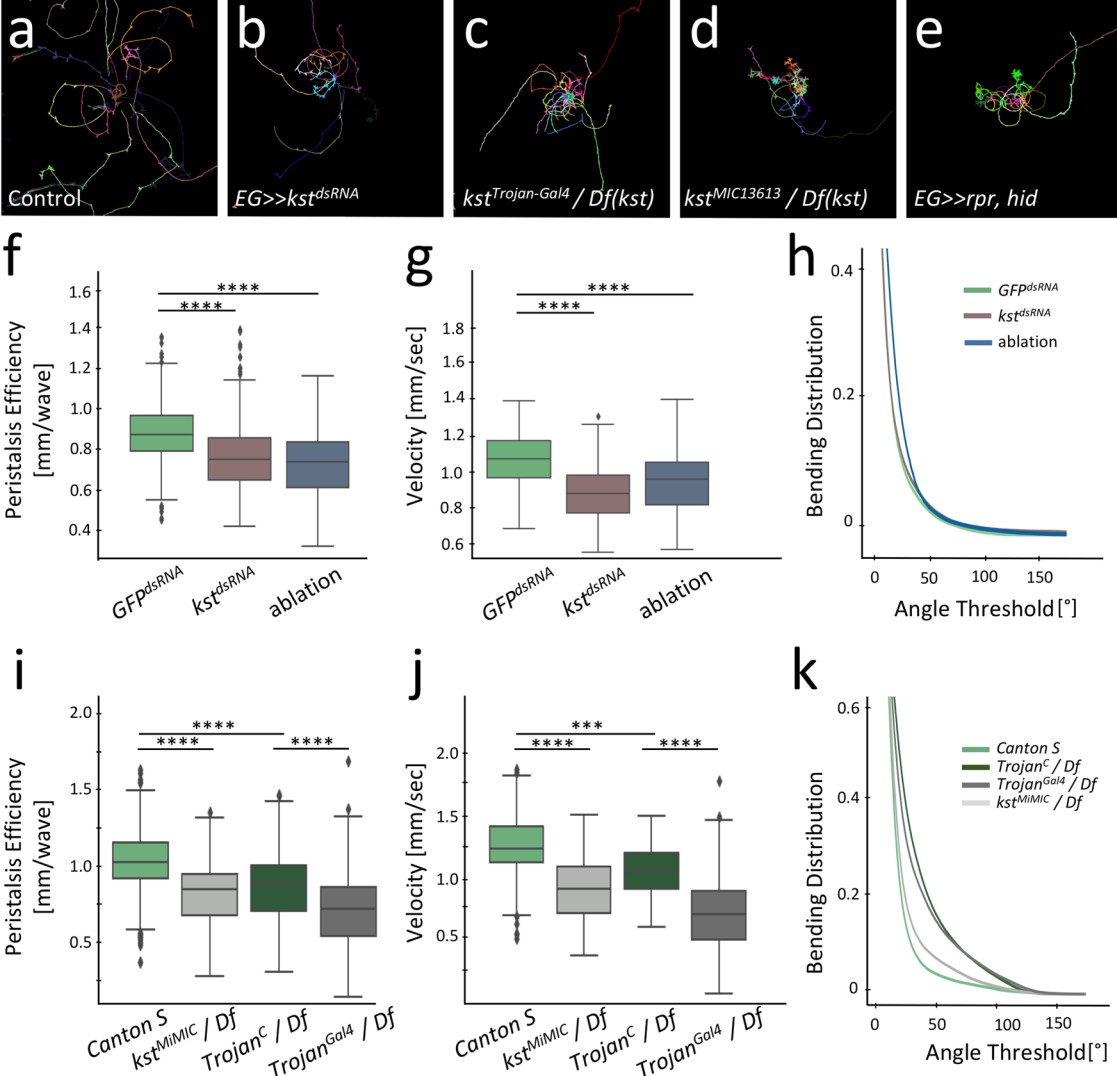

**Fig. 10 Karst is required in ensheathing glia for normal locomotor behavior. a** Exemplary locomotion tracks of control wandering third instar larvae with the genotype [*83E12-Gal4*^AD^, *repo-Gal4*^DBD^, *UAS-GFP*^dsRNA^]. Larvae move in long run phases. **b** Upon *kst* knockdown in ensheathing glia (EG) only [*83E12-Gal4*^AD^, *repo-Gal4*^DBD^, *UAS-kst*^dsRNA^], larvae do not explore the tracking arena. **c** The insertion of a Trojan Gal4 element into the first coding exon of *kst-RE(RG)* generates an PE and PG isoform specific mutant (see Fig. 8). Larvae with the genotype [*kst*^TrojanGal4^ / *Df(3 L)ED208*] show a comparable locomotion phenotype as RNAi knockdown larvae. **d** Locomotion phenotype of zygotic *karst* null mutant larvae with the genotype [*kst*^MiMIC13613^ / *Df(3 L)ED208*]. **e** Locomotion phenotype observed following specific ablation of the ensheathing glia [*UAS-rpr; 83E12-Gal4*^AD^; *repo-Gal4*^DBD^, *UAS-hid*]. **f–h** Quantification of peristalsis efficiency, velocity and bending distribution between control larvae, *kst* knockdown and ensheathing glia ablated animals. Box plots show median (line), boxes represent the first and third percentiles, whiskers show standard deviation, diamonds indicate outliers. **i–k** Quantification of the same parameters comparing wild type Canton S with *kst* null larvae and a *kst*^Trojan^ control strain that carries the Trojan insertion in a non-productive orientation [*kst*^MiMIC03134-Trojan-C^ in trans to the *Df(3 L)ED208*] with the isoform specific *kst*^PE, PG^ mutant [*kst*^MiMIC03134-TrojanGal4^ / *Df(3 L)ED208*. Quantification Wilcoxon rank-sum test, two tailed, $n = 50$. Peristalsis Efficiency [mm/wave]: GFP^dsRNA^ – kst^dsRNA^: ****$p = 2.65E-32$ GFP^dsRNA^ – ablation; ****$p = 3.22E-30$; velocity [mm/sec]: GFP^dsRNA^ – kst^dsRNA^: **** $p = 3.81E-11$; GFP^dsRNA^ – ablation: ****$p = 1.54E-05$; Peristalsis Efficiency [mm/wave]: Canton S – kst^MiMIC^ / Df: ****$p = 3.98E-20$; Canton S – Trojan ^C^/ Df: ****$p = 4.61E-13$; Trojan^C^ – Trojan^Gal4^ / Df: ****$p = 1.22E-13$; velocity [mm/sec]: Canton S – kst^MiMIC^ / Df: ****$p = 1.15E-06$; Canton S – Trojan^C^/ Df: ****$p = 1.21E-03$; Trojan^C^ – Trojan^Gal4^ / Df: ****$p = 5.94E-13$. Source data are provided as a Source Data file.

also occurs between cortex and ensheathing glia[20]. Such interactions might also contribute to the functional differences that appear likely to be associated with for example dorsal or ventral ensheathing glial cells. Unfortunately, no specific molecular tools exist that allow to test this hypothesis.

A likely role of ensheathing glial cells is to establish a diffusion barrier around the neuropil, as was demonstrated by our dye injection experiments in ensheathing glia ablated larvae. However, we cannot exclude the possibility that ensheathing glia instruct neighboring cells such as the cortex glia or astrocyte-like glia to form a barrier around the neuropil. The separation of the

neuropil from the cortex in the larva might facilitate the formation of signaling compartments specified by astrocyte-like cells. The 10-fold increase in the number of ensheathing glial cells in the adult brain is likely to further organize neuronal functions into discrete domains[35,78–80].

As also observed in vertebrates, the formation of signaling compartments can be detected on a gross anatomical level as well as on a circuit level. In both cases, the neuropil-associated glia comprising ensheathing glia are at center stage[25] and complexity of regional compartmentalization increases in the adult, mirroring the increase in ensheathing glial cell number[33,34,81]. In

vertebrates, a classical example for this fundamental principle can be seen at the rhombomeric organization of the vertebrate hindbrain[82]. Here, glial cells provide essential functions in boundary formation, too[83]. Thus, glia may not only structure the nervous system by compartmentalizing synapses but also act on the level of larger functional units.

The establishment and maintenance of cell polarity is a fundamental cell biological feature and is required for the development of different cell types and tissues. For instance, the mammalian nervous system is protected by the blood–brain–barrier established by polar endothelial cells whereas the blood–brain barrier around the invertebrate nervous system is comprised of polarized glial cells[4,13,84–86]. Cell polarity is induced by evolutionarily conserved mechanisms[87,88]. Apical polarity regulators (APRs) comprise a diverse set of proteins centered around the two apical polarity protein complexes Bazooka/Par3 and Crumbs[89]. In addition, basolateral regulators including the Scribble complex (Scribble, Discs large and Lethal (2) giant larvae and the kinases PAR-1 and LKB1 participate in the definition of a polar cell phenotype[90]. Some of these genes (Par1, Scrib, dlg1) appear to be expressed by adult ensheathing glia[11]. For Scrib but not Par1 or dlg1 this can be confirmed protein trap insertion lines. However, RNAi mediated knock-down of these genes did not cause any morphological defects in larval ensheathing glia. Possibly, similar as in follicle cell devel-opment, different mechanisms are in place to guarantee the establishment and maintenance of the polar ensheathing glial cell phenotype[91].

A hallmark of polarized cells is the asymmetric distribution of lipids with $PIP_2$ being enriched at the apical surface[62] and reduction in $\beta_H$-Spectrin causes a disrupted $PIP_2$ localization. This suggests, that a polar spectrin cytoskeleton might orchestrate lipid composition at the apical membrane domain which subse-quently affects polarized distribution of transmembrane proteins such as the $Na^+/K^+$ ATPase. This might cause to a differential distribution of $Na^+$ ions which are used by antiporters involved in metabolite transport. One of them is the Excitatory amino acid transporter 2 (EAAT2)[92]. Antibodies against EAAT2 are available[27] but unfortunately do not allow the analysis of its subcellular localization in larvae. Together, the different transport systems expressed in the ensheathing glia will also affect extracellular environment and thus diffusion properties in and out of the neuropil, which needs to be addressed in further studies.

The synaptic neuropil is localized at a unique position within the nerve cord very close to the dorsal surface and thus, the hemolymph. Whereas the ventral as well as the lateral parts of the neuropil are flanked by neuronal cell bodies embedded in cortex glia, only few neuronal cell bodies are found at the dorsal surface of the nervous system. More importantly, no cortex glial cells are found dorsally to the neuropil[20,28]. Thus, metabolites can be transported from the blood–brain barrier to the neuropil only by the ensheathing glia. Upon ablation of the ensheathing glia, the blood–brain barrier forming glial cells might provide enough metabolites to ensure an almost normal function of the nervous system. In case of suppression of Nrv2 expression in the ensheathing glia, ion gradients required for directed transport are not established across the ensheathing glial plasma membrane. At the same time the barrier provided by the ensheathing glia is still in place and therefore metabolically isolates the neuropil which results in an increased mobility possibly aimed to direct the larva to new and better food sources. The neuropil itself is highly organized and its dorsal domain is dedicated to synaptic con-nections of motor neurons[93]. Possibly, the motor neurons are more sensitive to nutrient supply and thus a more direct meta-bolic support through the ensheathing glia is required.

## Methods

**Drosophila genetics**. Flies were raised on food prepared according to a recipe of the Bloomington stock center (https://bdsc.indiana.edu/information/recipes/bloomfood.html). Depending on the culture conditions, fly stocks were flipped every 2–4 weeks. Experimental flies were always kept at 25 °C. For crosses flies were anesthetized with $CO_2$. Flies of undesired genotypes were either inactivated by placing them at −30 °C for 2 days or by placing them in 70% ethanol as approved by the authorities.

The following flies were obtained from public stock centers: 83E12-Gal4 (BDSC#40363), 83E12-LexA (BDSC #54288), P{y[+t7.7] w[+mC]= 13XLexAop2-mCD8::GFP}attP2 (BDSC#32203), MCFO-2: pBPhsFlp2::PEST;; HA_V5_FLAG_OLLAS (BDSC#64086), string^dsRNA (NIF-fly, 1395R-2), P{w[+mC]=UAS-htl.lambda\cI.M}40-22-2 (Kyoto Stock collection, 108191), P{w[+mC]=UAS-Lam.GFP}3-3 (BDSC#7376), P{w[+mC]=UAS-mCD8::GFP.L} LL6 (BDSC#5130), P{UAS-mCD8.ChRFP}2 (BDSC#27391), P{w[+mC]=UAS-rpr.C}27 (BDSC#5823), P{w[+mC]=UAS-hid.Z}2/CyO (BDSC#65403), P{4 C.UAS-PLCδ-PH-mCherry} (BDSC#51658), PBac{806.LOX-SVS-2}if^CPTI004152 (#4152), Mi{PT-GFSTF.0}trol^MI04580-GFSTF.0 (BDSC#60214), Mi{PT-GFSTF.1}kst^MI03134-GFSTF.1 (BDSC#60193), Mi{MIC}kst^MI03134 (BDSC#36203), Mi{MIC}kst^MI13613 (BDSC#59172), P{XP}kst^d11183 (BDSC#19345), kst^P{EPgy2}EY01010 (BDSC#15488), kst^dsRNA (BDSC#50536), Df(3 L)ED208 (BDSC#8059), Mi{PT-GFSTF.1}nrv2^MI03354-GFSTF.1 (BDSC#59407), nrv2^dsRNA (VDRC#2260), P{w[+mC]=lexAop-rCD2.RFP}2; P{w[+mC]=UAS-CD4-spGFP1-10}3, P{w[+mC]=lexAop-CD4-spGFP11}3 (BDSC#58755), 55B12-Gal4 (BDSC#39103), VGlut[OK371]-Gal4 (BDSC#26160), GFP^dsRNA (BDSC#9331). All other fly stocks were generated or were provided by other labs: 83E12-Gal4^AD, repo -Gal4^DBD, Mi{Trojan-GAL4.0} kst^MI03134, dally::GFP, viking::GFP (all this study), PBac{806.LOX-SVS-2} if^CPTI004152 (FlyProt, 4152), UAS-fzr^53, UAS-PH-Akt::GFP (S. Luschnig, Münster, Germany). The MCFO flies were heat-shocked for 1 h at 37 °C in a water bath at different developmental time points as indicated. The Trojan-Gal4 cassettes was exchanged by phiC31-integration protocols[94].

**Immunohistochemistry**. Immunohistochemistry for larval, pupal and adult brains was performed using standard protocols[95]. The following antibodies were used: anti-Repo ([1:5], Cat# 8D12, RRID: AB_528448, Developmental Studies Hybri-doma bank); anti-mCherry ([1:1,000], Cat# M11217, RRID: AB_2536611); anti-V5 ([1:1,000], Cat# R960-25, RRID: AB_2556564) and DAPI ([1:1,000], Cat# D1306, RRID: AB_2307445) (all from Invitrogen™); anti-Flag ([1:10], Cat# SAB4200071, RRID: AB_10603396, Sigma-Aldrich); anti-Ollas ([1:1,000], Cat# MAB7372, RRID: AB_11152481, Abnova), anti-HA ([1:500], Cat# 9015116, RRID: AB_2820200, BioLegend); anti-GFP ([1:1,000], Cat# A-6455, RRID: AB_221570, Life Technol-ogies); Anti-phospho-histone H3 (phospho S28) ([1:500], RRID:AB_2295065, Cat# ab10543, Abcam); anti-HRP-Alexa Flour 647 conjugated ([1:1,000], RRID: AB_2338952, Cat# 123-005-021, Dianova); anti-Nazgul ([1:250], gift from B. Altenhein, Cologne). Rabbit anti-Rumpel [1:500] (gift of K. Yildirim, Münster). A guinea Pig anti-serum generated against a GST-EAAT2 fusion protein [1:500][27] was a gift of D. van Meyel. A C-terminally located peptide (^3622LADERR-RAEKQHEHRQN^3639) shared by all ß_H-Spectrin proteins was used to immunize rabbits (Pineda, Berlin). ß_H-Spectrin was used in a dilution of [1:5,000]. For confocal imaging and analysis samples were imaged using a Zeiss LSM 880 with Zeiss imager ZEN 2.3 or a Leica SP8 confocal microscope using Leica Leica Application Suite X (LAS X) 3.5.7.23225 and Imaris 8.4.1 Oxford instruments software for 3D reconstruction.

**Electron microscopic analysis**. All ssTEM data were generated at the Howard Hughes Medical Institute Janelia Research Campus and processed as described[31,43]. The CATMAID interface[44,96] was used to annotate glial cells associated with the neuropil. Glial cell bodies were identified by characteristic color of the cytoplasm and non-neuronal morphology. A cell body node was positioned at the glial soma and annotated using the following information: hemineuromere, glial subtype, serial number for cells in one hemineuromere. The tools built into CATMAID were used to generate 3D renderings and extract image data. Cells identified in RAW data were highlighted using Adobe Photoshop CS6.

**Measurement of DAPI intensity**. To determine the polyploidy of ensheathing glia, we stained larval and adult brains with DAPI, Repo and Elav in order to identify glial cells and neurons. We compared DAPI stained nuclei identified by 83E12-Gal4 > UAS-Lam::GFP with DAPI staining of directly neighboring neuronal nuclei identified by anti-Elav staining. The amount of DAPI staining of neighboring glial and neuronal nuclei was determined using Fiji 1.52b[97]. In each case the entire nuclear volume was measured.

**EdU incorporation assay**. First instar larvae were placed on freshly prepared fly food containing 0.2 mM EdU and were kept on the food at 25 °C until the third instar stage. Brains of the desired age were dissected and stained for EdU presence using the Click-iT Plus EdU detection kit of Thermo Fisher (# C10638) according to the instructions of the manufacture.

**Dye injection assay**. Brains of wandering L3 larvae were dissected on ice and immediately placed on Poly-L-lysine coated object slide and covered by 10 S Volatef halocarbon oil. An automated injection station (FemtoJet 4 L, Eppendorf) was used to inject 2.5 mM 10 kDa Texas-Red conjugated dextran (Cat#D1863, Invitrogen) diluted in $H_2O$ into the neuropil of one brain lobe in each brain. For injections, we used borosilicate glass capillaries of 1 mm outer diameter and 0.58 mm inner diameter (Harvard apparatus 30-0016 capillaries GC100-10) that were pulled on a P-1000 Sutter instrument micropipette puller to reach a tip diameter of about 10 μm as commonly used for DNA injections.

**Longevity assay**. Flies were raised at 25 °C. The offspring of the control and the ablation experiment were collected for up to 3 days immediately after hatching and separated between males and females. 20 mated females were put in a vial and a total of 200 animals were observed over a period of time. The living flies were counted and placed on fresh food without anesthesia every two days.

**Larval locomotion analysis**. Larvae for behavioral experiments were raised at 25 °C according. Larval locomotion of wandering third instar larvae was tracked by the FTIR-based Imaging Method (FIM) based on frustrated total internal reflection (FTIR) at room temperature[74,75]. 10-15 larvae per genotype were placed on the tracking arena and after a 60 s accommodation phase recording started for 3 minutes using FIMtrack v3.1.28.5 and analyzed suing FIMAnalytics v0.1.1.2.

**Quantification, statistical analysis and reproducibility**. All confocal analyses were repeated independently at least three times with at least 5 specimens each and similar results were obtained. Fiji was used to investigate polar distribution. Cells were divided in two ROIs and the fluorescence intensity ratio between apical and basal was compared. Counting of the nuclei and the astrocytes was performed in a 3D model in Imaris 8.4.1 Oxford Instruments. Fluorescence intensities for the Dextran Assay were measured using Fiji 1.52b[97] using the normalized fluorescence intensity and the ratio between neuropil and cortex. Statistical analysis and calculation were obtained by Prism 6.0 and Excel 2016.

**Reporting summary**. Further information on research design is available in the Nature Research Reporting Summary linked to this article.

## Data availability

The fly stocks and datasets generated during and analyzed during the current study are available from the corresponding author on reasonable request. Source data are provided with this paper. The processed single-cell RNA-sequencing data in Supplementary Fig. 6 was generated from publicly available data[11]. The raw data is stored in a loom format (https://github.com/linnarsson-lab/loompy) and visualized by SCope (http://scope.aertslab.org). Source data are provided with this paper.

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

## Acknowledgements

We are grateful to B. Altenhein, S. Luschnig, E. Peco and D. van Meyel, and K. Yildirim for generously providing us with antibodies and flies. E. Contreras, M. Krahn, R. Stanewsky, S. Luschnig, S. Schirmeier and S. Rumpf for comments on the manuscript. We thank the Fly EM Project Team at HHMI Janelia and Richard D. Fetter for the *Drosophila* larval CNS ssTEM volume. We are grateful for all the support of all lab members. This work was supported by the Deutsche Forschungsgemeinschaft through funds to C.K. (SFB 1348, B5).

## Author contributions

N.P. and C.K. designed all experiments. N.P. conducted all experiments. H.K. identified polar expression of Karst, performed initial experiments and together with A.C. analyzed the TEM volume, A.C. contributed data and software, S.Re. and S.Ro. generated constructs used in this study. N.P. and C.K. together with input from all authors wrote the paper.

## Funding

## Competing interests

The authors declare no competing interests.
