## [Peer Review File · Nature Communications]

Editorial Note: Parts of this Peer Review File have been redacted as indicated as we could not obtain permission to publish the reports of reviewer 2.

Reviewers' Comments:

Reviewer #1:

Remarks to the Author:

Glial cells play crucial roles in the developing and mature central nervous system. However, the development and function of the different classes of glia at work remain poorly understood. This study of Nicole Pogodalla and colleagues in Christian Klaembt's lab focuses on ensheathing glial cells in the larval and adult *Drosophila* ventral nerve cord. The study begins with a thorough assessment of numbers and emerging morphologies of this glial cell type during development, as well as its interactions with astrocytes. Furthermore, the study provides strong evidence that ensheathing glia are important for neuropil isolation by providing a diffusion barrier, that they are polarized with differential distributions of PIP2 and PIP3 and that beta heavy Spectrin (encoded by the gene *karst*) mediates interactions with PIP2 at the interface with the neuropil. Finally, alterations of ensheathing glia have functional impacts on larval locomotion behavior. This detailed study thus provides in my view new insights into the so far elusive role of ensheathing glia, which is of wide interest for glial biologists. Nevertheless, I would like to add some suggestions to strengthen this manuscript.

Specific comments :

1. Generally, the study would benefit from the addition of higher resolution images to underscore the points made. For instance on page 6, Figure 1, the development of thin lateral processes is described. However, changes are barely visible. Similarly, there is a detailed assessment of the size of DAPI labeled nuclei to distinguish polyploid and diploid ensheathing glia. Yet, in the provided images the differences are not visible.
2. Along these lines, the authors describe on page 8 that astrocyte extend more branches to compensate for the loss of ensheathing glia. However, the branches are barely visible in the provided figures. Points should be supported by high resolution images. Moreover, as there is no functional compensation, it would be an important and interesting addition to assess effects in different (dorsal and ventral) regions. In the literature, a 1:1 ratio of alternating ensheathing and astrocyte glia has been reported, as well as some interdependence in their formation/specification. How does this fit into the interpretation of presented data?
3. Page 7. The authors describe that expression of activated heartless in ensheathing glia increases proliferation during mid-pupal development. However, except for the increasing number of adult glia, evidence for proliferation at pupal stages has not been added. It would be important to either reformulate or assess proliferation patterns using BrdU/EdU during pupal development.
4. Page 10. It would be helpful to add to Figure 5A the location of the cortex, it was not clear immediately that the labeling indicates only the direction towards the blood brain barrier relative to the neuropil. Also, the underlying description of the quantification ratio BBB/neuropil appears unclear and does not allow the reader to follow the precise measurement.
5. On Pages 10 and 11, an entire section is dedicated to the description of ECM components, yet none of the data are shown in the main figures. The authors should move the most pertinent data to a main figure or shorten the section.
6. Page 14. Next, Beta H Spectrin is assessed using genetic markers and a novel antibody. Assessing the function of Spectrin, the authors mention that basal cellular protrusions around neurons are less evident and that there are more pronounced differences between the brain and the VNC. However in Figure 7, it is not quite clear for what to look, as the reductions seem to be variable and no similar controls are shown alongside. Moreover, the question remains how to untangle branch extension defects from effects on PIP sensor localization/effects.
7. The authors provide some insights into the function of *karst* using loss and knock-down approaches. Is it possible to complement these findings with GOF approaches for *karst* to strengthen the conclusions by an assessment of sufficiency.
8. In the discussion, the authors raise an interesting point about differences in ventral and dorsal ensheathing glia requirements. Would it be possible to assess these regional effects more in detail and also functionally?
9. Another suggestion would be to provide a summarizing diagram to facilitate the understanding of the differential interactions of *karst*, PIPs and their link to branch morphology.

Minor comments :

1. Page 2 (and other locations of the manuscript) : nrv = nervana
2. Page 3. Extreme seems un unusual choice of words and whether the invertebrate nervous system is a much simpler structured nervous system is perhaps debatable.
3. Page 4. Engulf does not seem to be the right wording.
4. Page 5. Eaat2 is likely not expressed in ensheathing glia to suppress sleep – this is only one effect or role.
5. Page 5. Line 98. It should be growth of astrocyte-like glia.
6. Page 6. Line 121. The pupal stage is not clear, especially because the authors do not refer to a pupal, but an adult CNS ?
7. Generally, arrows etc should feature in the figure legend rather the main text.

[Redacted]

Reviewer #3:

Remarks to the Author:

This manuscript provides a rich source of new information of the development, cell biology and function of the ensheathing glia in *Drosophila*. Cutting edge technology is utilized, the data are well described and documented, and the paper adds many new impactful data to the field.

Suggestions for improvement:

1.Results line 124ff: The developmental history of ensheathing glia has been studied at some level of detail for the brain (Omoto et al., 2014). Here, embryonically born primary ensheathing glia persist throughout the larval period without increase in number; they are replaced by secondary ensheathing glia that derive from the dorsomedian (DM) lineages. These data could be used as a "springboard" to ask if/how ensheathing glia development differs from that known for the brain. In this context it should be mentioned: what lineages of the VNC generate primary ensheathing glia? What is known about origin of secondaries in the VNC

2.Line 153: The developmental model proposed here assumes that the diploid ensheathing glia in the thoracic segments are differentiated but continue to divide. Is it clear that there are at the larval stage no secondary progenitors that proliferate, as in the brain? Or could it be that the diploid cells described here can be considered as progenitors, rather than differentiated glial cells?

3.Line 199: the barrier function of ensheathing glia seems to be only partial, correct? That could be stressed more; and compared for example with a more "stringent" barrier function, such as that of the surface glia (for these similar diffusion experiments have been done in the past, correct? Also: It could be discussed at the start of the barrier function whether or not ensheathing glia have septate junctions, which are the mediators of the "stringent" barrier at the brain surface.

4.Figure 5A: the schematic is not intuitive. I get the idea of apical vs basal, but what are hatched boxes with neuron? Is the localization of PIP2/3 not shown for the ensheathing glia? Why is the term blood brain barrier used for ensheathing glia? Isn't that the function of surface glia?

5.line 296ff: the karst phenotype: why are the authors emphasizing "cellular protrusions around neuronal cell bodies"? What happens to the processes around the neuropil? The essential structure of EGs I am familiar with is to form "horizontal" processes along the cortex-neuropil boundary. Are there widespread processes around somata as well? This is the main role of cortex glia. (Maybe it was stated in this MS at a previous section; I didn't go back to look for it. But in any case, for the reader following along, the sudden focus on basal processes of EG surrounding neurons deserves explicit mentioning)

6.In this context: in Fig.7 and others, the high mag panels showing cross sections focus on the dorsal cortex of the VNC. Here, the cortex is "unusual", very thin, with mainly the cell bodies of glia and a few neurons. Why not focusing on the lateral or ventral cortex, which also have a complete EG sheath?

7.Line 333ff: Taking the text literally, I assume that it is b-spectrin, and NOT bH-spectrin/Karst that is used for Nrv2 localization? If yes, this should be emphasized, because in the previous paragraph the authors talk about bH-spectrin function.

8.line 340ff: this goes too fast. In previous sections, the authors carefully documented the expression and structural phenotype of bH-spectrin/Karst; then they proceeded to the behavioral test. For b-spectrin, only the behavioral phenotype is mentioned, which, surprisingly I think, is the opposite of the bH-spectrin phenotype. What about structural changes in EGs?

9.In this context: being aware of the EG structure (sheath processes) and the previously reported experiment of a barrier function (Dextran), the final conclusion doesn't really relate to this: what is the relationship between barrier function and "provision of the neuropil with essential

metabolites”?

We are thankful for the time of the reviewers and their positive and helpful suggestions to improve our manuscript. We have now addressed all the issues raised by the reviewers and hope that no further questions remained open. The original review is printed in grey and our comments in black.

Reviewer #1 (Remarks to the Author):

Glial cells play crucial roles in the developing and mature central nervous system. However, the development and function of the different classes of glia at work remain poorly understood. This study of Nicole Pogodalla and colleagues in Christian Klaembt's lab focuses on ensheathing glial cells in the larval and adult *Drosophila* ventral nerve cord. The study begins with a thorough assessment of numbers and emerging morphologies of this glial cell type during development, as well as its interactions with astrocytes. Furthermore, the study provides strong evidence that ensheathing glia are important for neuropil isolation by providing a diffusion barrier, that they are polarized with differential distributions of PIP2 and PIP3 and that beta heavy Spectrin (encoded by the gene *karst*) mediates interactions with PIP2 at the interface with the neuropil. Finally, alterations of ensheathing glia have functional impacts on larval locomotion behavior. This detailed study thus provides in my view new insights into the so far elusive role of ensheathing glia, which is of wide interest for glial biologists. Nevertheless, I would like to add some suggestions to strengthen this manuscript.

We are thankful for this positive assessment of our manuscript and appreciate all of the below made comments. We changed the manuscript as explained in detail in the below sections.

1. Generally, the study would benefit from the addition of higher resolution images to underscore the points made. For instance, on page 6, Figure 1, the development of thin lateral processes is described. However, changes are barely visible.

We agree, and provided higher resolution images, including a better labelling. We also added a more thorough characterization of the developing ensheathing glia by including electron microscopic work which originated from an extensive collaboration with Albert Cardona, whom we now included in the author list. This led to a new Figure 2 (shown below) and a corresponding representation in the text. The TEM analysis also demonstrates that the ensheathing glia form an extensive membrane overlap in third instar larval stages which presumably contributes to the barrier properties of the ensheathing glia (See Figure 2 I,J).

New Figure 2: Larval development of the ensheathing glia.

Similarly, there is a detailed assessment of the size of DAPI labeled nuclei to distinguish polyploid and diploid ensheathing glia. Yet, in the provided images the differences are not visible.

We apologize for not clearly showing how we generated the data. An example is given below. We stained the nervous systems of the genotype 83E12-Gal4, UAS-lam::GFP with DAPI, anti-elav and anti-GFP. We then generated confocal image stacks and determined the DAPI fluorescence intensity of a ensheathing glial nucleus (defined by GFP expression) with a neighboring neuronal nucleus (defined by Elav expression). The below Figure shows two examples and randomly picked high power examples. Note that all images are taken from the confocal projection of all stacks - whereas the analysis of DAPI fluorescence was done from only the sections that

contained the respective nucleus. We have not added the Figure to the supplementary data but are absolutely willing to do so upon editors/reviewers advice.

Thoracic Glia/
Neuron DAPI ratio

1	1,3	1	1,4
2	1,0	2	1,4
3	1,0	3	1,3
4	0,8	4	1,5
5	1,0	5	1,2
6	1,9	6	1,9
7	1,8	7	1,4
8	1,5	8	1,4
9	1,8	9	1,1
10	1,5	10	1,6
11	1,5	11	1,6
12	1,3	12	1,2
13	1,4		

Abdominal Glia/
Neuron DAPI ratio

1	2,0	1	2,0
2	1,8	2	2,3
3	1,9	3	1,7
4	2,3	4	2,2
5	1,7	5	1,9
6	2,2	6	1,9
7	2,4	7	2,0
8	2,0	8	1,7
9	2,0	9	2,3
10	1,9	10	2,1

Figure demonstrating the quantification of DAPI staining in glial nuclei.

2. Along these lines, the authors describe on page 8 that astrocyte extend more branches to compensate for the loss of ensheathing glia. However, the branches are barely visible in the provided figures. Points should be supported by high resolution images.

We thank the reviewer for this suggestion. We have therefore included higher magnifications which clearly show that astrocytes extend large cell protrusions at the neuropil cortex interface in the absence of ensheathing glia. The changed Figure is shown below.

Part of the revised Figure 4 highlighting Highlighting the protrusions of the astrocyte-like glial cells.

Moreover, as there is no functional compensation, it would be an important and interesting addition to assess effects in different (dorsal and ventral) regions. In the literature, a 1:1 ratio of alternating ensheathing and astrocyte glia has been reported, as well as some interdependence in their formation/specification. How does this fit into the interpretation of presented data?

We agree and have discussed a possible interdependence of ensheathing and astrocyte-like glia in the new version of manuscript. The formation of ensheathing glia and astrocyte-like glial cells has been beautifully studied by Peco et al., 2016. The 83E12-Gal4 driver is activated after specification of ensheathing glia and no activity is found in astrocyte-like cells. Induced cell death by expression of hid using 83E12-Gal4 is thus not expected to interfere with the initial formation of astrocyte-like cells. The difference between dorsal and ventral ensheathing glial cells is of obvious interest, but this cannot be addressed currently. While these two glial subpopulations clearly exist, we have not found any Gal4 driver (or any related tools) that allows to discriminate between the two cell types.

3. Page 7. The authors describe that expression of activated heartless in ensheathing glia increases proliferation during mid-pupal development. However, except for the increasing number of adult glia, evidence for proliferation at pupal stages has not been added. It would be important to either reformulate or assess proliferation patterns using BrdU/EdU during pupal development

We agree with the reviewer and did the following.

We added the requested images showing the effect of activated heartless in pupal brains. We also revised and rephrased the entire section on cell proliferation, which is now more concise. This paragraph was included to show that some of larval ensheathing glia persist until adult stages and that some ensheathing glia can divide in pupal stages to generate adult ensheathing glia. As suggested by the reviewer we better assessed the proliferation patterns and performed anti phospho-histone H3 stainings to directly detect dividing glial cells (see below). These data are now summarized in a new supplementary Figure and hopefully sufficiently support our suggestion that 83E12-Gal4 positive ensheathing glia are able to divide.

New supplementary Figure 2: FGFR induced glial proliferation during pupal stages.

Part of a new supplementary Figure 3: Phospho-histone H3 staining during development.

We also performed EdU labeling experiments. We fed larvae after hatching on EdU containing food and analyzed the CNS of wandering third instar larvae. EdU incorporation was found in ensheathing glial cells (*83E12-Gal4* positive cells, see Figure below). Since staining with anti-phospho-histone H3 antibodies directly labels dividing cells, we did not include the EdU data in the manuscript.

Additional Figure for the reviewer: EdU is taken up by *83E12-Gal4*, *UAS-lam::GFP* expressing larvae,

4. Page 10. It would be helpful to add to Figure 5A the location of the cortex, it was not clear immediately that the labeling indicates only the direction towards the blood brain barrier relative to the neuropil. Also, the underlying description of the quantification ratio BBB/neuropil appears unclear and does not allow the reader to follow the precise measurement.

We agree that the schematic image was misleading. We therefore replaced it by two new schematic figures. One shows an overview of a cross section of the ventral nerve cord to demonstrate the overall organization and the location of the neuropil close to the dorsal blood-brain barrier and importantly also indicates the cortex glia, which is not found in the dorsal ventral nerve cord. This schematic view on the ventral nerve cord is added as Figure 1J.

The second schematic drawing shows the relation of the ensheathing glia, the blood-brain barrier and the neuropil and is now included in the revised Figure 6G. In addition, we added two new data images showing the localization of the cortex glia. We used *83E12-LexA*, and used this to independently modify ensheathing glia and cortex glia. We expressed GFP in the ensheathing glia using *83E12-LexA*

and expressed Cherry in the cortex glia using *55B12-Gal4*. Furthermore, the new 83E12-LexA transgene allowed to perform GRASP (GFP reconstitution across synaptic partners) experiments to label the interface between cortex glia and ensheathing glia. Following expression of a membrane tethered GFP fragment (GFP^{11}) in ensheathing glia and expression of the complementing portion of GFP (GFP^{1-10}) in cortex glia [*55B12-Gal4*, *83E12-LexA*, *UAS-GFP1-10*, *LexAop-GFP11*], we note a reconstitution of fluorescence only at the lateral and ventral cortex neuropil interface. Finally, we used *OK371-Gal4* to drive expression of Cherry in all glutamatergic neurons together with 83E12-LexA driven GFP expression. This clearly labels some of the dorsal neurons engulfed by the ensheathing glia. The data are now presented as a new supplementary Figure (see below).

New supplementary Figure 1.

5. On Pages 10 and 11, an entire section is dedicated to the description of ECM components, yet none of the data are shown in the main figures. The authors should move the most pertinent data to a main figure or shorten the section.

Thank you for this suggestion. We have now moved the data Figure to the main text where it is now Figure 7 and at the same time shortened the text.

6. Page 14. Next, Beta H Spectrin is assessed using genetic markers and a novel antibody. Assessing the function of Spectrin, the authors mention that basal cellular protrusions around neurons are less evident and that there are more pronounced differences between the brain and the VNC. However, in Figure 7, it is not quite clear for what to look, as the reductions seem to be variable and no similar controls are shown alongside. Moreover, the question remains how to untangle branch extension defects from effects on PIP sensor localization/effects.

This is a very good point and we invested more energy in dissecting the mutant *karst* phenotype. We revised former Figure 7 which is now Figure 9 and included a control in addition to showing better images of the *karst* mutant. Here, a lack of basal protrusions can be seen. In addition, we show the distribution of the Nervana2 protein in *karst* mutant ensheathing glia. Together the data demonstrate that in the complete absence of *karst*, the ensheathing glial cells fail to extend basal processes. The notion that *karst* affects cell polarity is supported by *karst* knockdown experiments. Here, ensheathing glial cell morphology is less strongly affected compared to *karst* null mutants. Basal extensions can still be detected but the clear segregation of PIP2 and PIP3 in apical and basolateral plasma membrane domains is lost. Similarly, the polarized localization of the Nrv2 protein is affected.

New Figure 9.

7. The authors provide some insights into the function of karst using loss and knock-down approaches. Is it possible to complement these findings with GOF approaches for karst to strengthen the conclusions by an assessment of sufficiency.

We are thankful for this very good suggestion. We were not able to generate a *UAS-karst* transgene due to lacking full length cDNAs (the open reading frame encompasses 13 kb). To nevertheless achieve ectopic expression of *karst* we used an EP-insertion into the *karst* promoter region. *karst* overexpression causes the frequent formation of sponge-like, hyperconvoluted morphologies. Interestingly, the localization of both PIP₂ / PIP₃ and Nrv2 is affected. The data are now added to Figure 9.

8. In the discussion, the authors raise an interesting point about differences in ventral and dorsal ensheathing glia requirements. Would it be possible to assess these regional effects more in detail and also functionally?

This is a very good suggestion, but as we wrote above, unfortunately there are no tools presently available to discriminate the two ensheathing glial cell populations.

9. Another suggestion would be to provide a summarizing diagram to facilitate the understanding of the differential interactions of *karst*, PIPs and their link to branch morphology.

This is again an excellent idea and we now include a summary scheme (see above) where all the relevant aspects of the manuscript are summarized and hope that this Figure is helpful. In addition, we added a new graphical abstract.

Minor comments :

1. Page 2 (and other locations of the manuscript) : nrv = nervana
2. Page 3. Extreme seems an unusual choice of words and whether the invertebrate nervous system is a much simpler structured nervous system is perhaps debatable.
3. Page 4. Engulf does not seem to be the right wording.
4. Page 5. *Eaat2* is likely not expressed in ensheathing glia to suppress sleep – this is only one effect or role.
5. Page 5. Line 98. It should be growth of astrocyte-like glia.
6. Page 6. Line 121. The pupal stage is not clear, especially because the authors do not refer to a pupal, but an adult CNS ?
7. Generally, arrows etc should feature in the figure legend rather the main text.

All these minor issues were corrected.

[Redacted]

Reviewer #3 (Remarks to the Author):

This manuscript provides a rich source of new information of the development, cell biology and function of the ensheathing glia in *Drosophila*. Cutting edge technology is utilized, the data are well described and documented, and the paper adds many new impactful data to the field.

We are thankful for this very positive assessment of our manuscript. During the revision we added even more information including phospho-histone H3 staining, EdU incorporation, a new *83E12-LexA* transgene and a TEM reconstruction analysis.

Suggestions for improvement:

1. Results line 124ff: The developmental history of ensheathing glia has been studied at some level of detail for the brain (Omoto et al., 2014). Here, embryonically born primary ensheathing glia persist throughout the larval period without increase in number; they are replaced by secondary ensheathing glia that derive from the dorsomedian (DM) lineages. These data could be used as a “springboard” to ask if/how ensheathing glia development differs from that known for the brain. In this context it should be mentioned: what lineages of the VNC generate primary ensheathing glia? What is known about origin of secondaries in the VNC.

We are thankful for this suggestion and better clarified the difference in VNC and brain ensheathing glia. Further MCFO2 experiments indicated that larval ensheathing glia can persist to adult stages using MCFO2 experiments. Some diploid ensheathing glia found in thoracic neuromeres can indeed divide to generate at least some of the adult ensheathing glia, which is supported by phospho-histone H3 stainings. In conclusion, we show that some larval thoracic ensheathing glial cells are diploid and are able to divide to contribute to some ensheathing glial cells of the adult brain. The remaining ensheathing glial likely originate from secondary lineages. We want,

however, to stress that the focus of this manuscript is the function of the ensheathing glia as internal diffusion barrier across the neuropil, their polarized cell morphology and the identification of genes required for the establishment of this polarized internal glial barrier.

2.Line 153: The developmental model proposed here assumes that the diploid ensheathing glia in the thoracic segments are differentiated but continue to divide. Is it clear that there are at the larval stage no secondary progenitors that proliferate, as in the brain? Or could it be that the diploid cells described here can be considered as progenitors, rather than differentiated glial cells?

These are excellent questions. The analysis of the L1 and the L3 TEM volume that is now presented in the paper suggests that all ensheathing glial cells are differentiated. This would be an indication that glial cells dedifferentiate in order to divide during pupal stages.

3.Line 199: the barrier function of ensheathing glia seems to be only partial, correct? That could be stressed more; and compared for example with a more “stringent” barrier function, such as that of the surface glia (for these similar diffusion experiments have been done in the past, correct? Also: It could be discussed at the start of the barrier function whether or not ensheathing glia have septate junctions, which are the mediators of the “stringent” barrier at the brain surface.

The reviewer addresses an important point which we should have explained better in the first version of the paper. So far, dye penetration experiments have not been performed in the larval CNS. Instead, labelled dextran was injected into the hemolymph of late embryos or into the hemolymph of adult flies. As stated above we injected the dye directly into the neuropil and determined the diffusion out of the neuropil. A direct comparison between the BBB and the neuropil barrier is thus difficult. In a different project, we addressed the third instar larval BBB and showed that the subperineurial glia covering the CNS do not allow penetration of labelled 70 kDa dextran (Winkler et al., submitted, *(the relevant Figure from the paper is provided below, similar data were obtained for 10kDa dextran)*). In wild type, larvae septate junctions formed between subperineurial glia mediate paracellular diffusion across the blood-brain barrier (Stork et al., 2008). However, even in the absence of septate junctions paracellular diffusion can be efficiently blocked by interdigitations of neighboring subperineurial glial cells (Babatz et al., 2018). To determine whether septate junctions also exist between ensheathing glial cells, we performed an electron microscopic analysis. As already suggested by confocal analysis, ensheathing glia start their differentiation around the neuropil already in first instar larvae. In a wandering third instar larva, glial processes are now fully engulfing the

neuropil. Importantly, ensheathing glia processes overlap extensively which increases the diffusion path at the boundary between CNS cortex and neuropil.

Figure: The blood brain barrier stays intact during larval and early pupal stages. Winkler et al submitted.

Third instar larvae or pupa of the indicated age were carefully opened to ensure no injury of the nervous system. 70 kDa fluorescent labelled dextran was added and diffusion into the nervous system was monitored using a confocal microscope.

4. Figure 5A: the schematic is not intuitive. I get the idea of apical vs basal, but what are hatched boxes with neuron? Is the localization of PIP2/3 not shown for the ensheathing glia? Why is the term blood brain barrier used for ensheathing glia? Isn't that the function of surface glia?

We apologize this schematic Figure was not self-explanatory enough. We therefore completely revised the Figure and also added new data on the distribution of the cortex glia and their contacts with the ensheathing glia by performing split-GFP experiments. See above. We also designed a graphical abstract which hopefully allows to more easily grasp the essence of our study.

Graphical abstract.

5.line 296ff: the karst phenotype: why are the authors emphasizing “cellular protrusions around neuronal cell bodies”? What happens to the processes around the neuropil? The essential structure of EGs I am familiar with is to form “horizontal” processes along the cortex-neuropil boundary. Are there widespread processes around somata as well? This is the main role of cortex glia.

This is a very good point. We had previously mentioned it in the text but did not stress it properly. There are no cortex glial cells at the dorsal part of the CNS and the ensheathing glial cells take over their part. Therefore, we added new data on the distribution of the cortex glia and their contacts with the ensheathing glia by performing split-GFP experiments. Please see our response to reviewer 1.

(Maybe it was stated in this MS at a previous section; I didn’t go back to look for it. But in any case, for the reader following along, the sudden focus on basal processes of EG surrounding neurons deserves explicit mentioning)

Please see above.

6.In this context: in Fig.7 and others, the high mag panels showing cross sections focus on the dorsal cortex of the VNC. Here, the cortex is “unusual”, very thin, with mainly the cell bodies of glia and a few neurons. Why not focusing on the lateral or ventral cortex, which also have a complete EG sheath?

The imaging of the very thin ensheathing glia at the lateral or ventral parts of the neuropil is difficult (see TEM image now provided in Figure 2). We were therefore not able to unambiguously discriminate between apical and basolateral domains of the ensheathing glia in these parts of the neuropil. The extensions of the dorsal

ensheathing glia around neuronal cell bodies and astrocyte-like glial cell bodies thus provide a fortunate possibility allowing the dissection of the polar cell phenotype of the ensheathing glia.

7.Line 333ff: Taking the text literally, I assume that it is b-spectrin, and NOT bH-spectrin/Karst that is used for Nrv2 localization? If yes, this should be emphasized, because in the previous paragraph the authors talk about bH-spectrin function.

We agree. We excluded beta-spectrin from the analysis and rephrased the entire paragraph.

8.line 340ff: this goes too fast. In previous sections, the authors carefully documented the expression and structural phenotype of bH-spectrin/Karst; then they proceeded to the behavioral test. For b-spectrin, only the behavioral phenotype is mentioned, which, surprisingly I think, is the opposite of the bH-spectrin phenotype. What about structural changes in EGs?

We agree and removed the information on β -spectrin. The last part of the manuscript now reads:

The above data show that larval ensheathing glia are polarized, ECM abutting cells that separate the neuropil from the CNS cortex and are required for longevity of the adult fly. To test whether the ensheathing glial cells are required for normal locomotor control during larval stages, we compared locomotion of control animals with those lacking ensheathing glia or with those with reduced β_{H} -Spectrin or Nrv2 expression using FIM imaging^{73,74}.

Control animals move on long paths interrupted by short reorientation phases that are characterized by increased body bending (Figure 10A). *karst* knockdown specifically in ensheathing glial cells [*83E12-Gal4AD*, *repo-Gal4DBD*, *UAS-kst^{dsRNA}*] causes a reduction in the peristalsis efficiency during go-phases (Figure 10A,B,F,G). Likewise, crawling velocity is reduced significantly (Figure 10G). This suggests that the specific lack of β_{H} -Spectrin in ensheathing glia causes a strong locomotor phenotype. We next analyzed mutant larvae to further validate the RNAi-induced phenotype. The Trojan-Gal4 insertion in the *karst*^{MiMIC03134} insertion is expected to affect only isoforms Karst-PE and Karst-PG (Figure 8A). As control, we used an insertion of the Gal4 element in the opposite, unproductive orientation. Similar as detected for the *kst* knockdown, we noted a decreased peristalsis efficiency and a reduced crawling velocity (Figure 10C,I,J). Larvae completely lacking zygotic *karst* expression [*karst*^{MiMIC13613} / *Df(3L)ED2083*] show a comparable larval locomotion phenotype (Figure 10D,I,J). This larval locomotion phenotype is similar to the one observed following ablation of the ensheathing glial cells using the genotype [*83E12-Gal4AD*, *repo-Gal4DBD*, *UAS-hid*, *UAS-rpr*] (Figure 10E). Thus, we conclude that polarized ensheathing glia that connect the blood-brain barrier with the dorsal neuropil are required for normal locomotor behavior. In order to perform vectorial transport, the Na⁺/K⁺ ATPase must act in a polarized fashion. We

thus also compared larval locomotion of animals with reduced *nrv2* expression to control animals. Interestingly, larvae with ensheathing glial cells lacking *nrv2* expression behave opposite to larvae that lack β_H -spectrin showing an increased peristalsis efficiency as well as an increased crawling velocity (supplementary Figure 8). The analysis of the role of polarized Nrv2 distribution for ensheathing glia physiology will thus be an interesting topic for future research.

9. In this context: being aware of the EG structure (sheath processes) and the previously reported experiment of a barrier function (Dextran), the final conclusion doesn't really relate to this: what is the relationship between barrier function and "provision of the neuropil with essential metabolites"?

Please see the above comment.

We are thankful for the time and the many excellent suggestions of the reviewers that helped to further improve our manuscript. We are confident that we responded to all questions and hope that all reviewers find the work now publishable.

References used in the text:

- Babatz, F., Naffin, E. and Klämbt, C.** (2018). The Drosophila Blood-Brain Barrier Adapts to Cell Growth by Unfolding of Pre-existing Septate Junctions. *Developmental Cell* **47**, 697–710.e3.
- Davie, K., Janssens, J., Koldere, D., De Waegeneer, M., Pech, U., Kreft, Ł., Aibar, S., Makhzami, S., Christiaens, V., Bravo González-Blas, C., et al.** (2018). A Single-Cell Transcriptome Atlas of the Aging Drosophila Brain. *Cell* **174**, 982–998.e20.
- Otto, N., Marelja, Z., Schoofs, A., Kranenburg, H., Bittern, J., Yildirim, K., Berh, D., Bethke, M., Thomas, S., Rode, S., et al.** (2018). The sulfite oxidase Shopper controls neuronal activity by regulating glutamate homeostasis in Drosophila ensheathing glia. *Nat Commun* **9**, 3514.
- Peco, E., Davla, S., Camp, D., M Stacey, S., Landgraf, M. and van Meyel, D. J.** (2016). Drosophila astrocytes cover specific territories of the CNS neuropil and are instructed to differentiate by Prospero, a key effector of Notch. *Development* **143**, 1170–1181.
- Stork, T., Engelen, D., Krudewig, A., Silies, M., Bainton, R. J. and Klämbt, C.** (2008). Organization and function of the blood-brain barrier in Drosophila. *Journal of Neuroscience* **28**, 587–597.

Reviewers' Comments:

Reviewer #1:

Remarks to the Author:

In this revised version of their manuscript, Nicole Pododolla, Christian Klaembt and colleagues have addressed all my earlier concerns, and added even more high-quality data than I would have expected, in particular the new TEM analysis. The images are now clear, shown at high magnification and provide strong evidence for all conclusions, presented in a concise manner. The study sheds light on the development and function of a glial cell type, the ensheathing glia, which we do not understand well so far. Their development and interactions with neurons and other glial cells, the role of polarity and the underlying molecular mechanisms, as well as the functional consequences of their disruption will be of high interest to the glial field in general. I look forward to seeing this study in print.

Minor comments:

Two little errors may need correction:

Line 88. "Likewise, the Excitatory amino acid transporter 2 ... is expressed by ensheathing glia and suppress sleep in the adult." is still not quite right.

Line 240. "Likewise, in when we ablated..." in should be removed.

[Redacted]

Reviewer #3:

Remarks to the Author:

The authors have done an outstanding job to respond in detail to the reviewers' comments. The revised manuscript is acceptable for publication.

Below please find a point to point reply to all reviewers. The concerns of the reviewer #2 are separated according to the different arguments. The text of the reviewers is printed in blue, our comments are in black. All reviewer comments are included here.

Reviewer #1 (Remarks to the Author):

In this revised version of their manuscript, Nicole Pododolla, Christian Klaembt and colleagues have addressed all my earlier concerns r, and added even more high-quality data than I would have expected, in particular the new TEM analysis. The images are now clear, shown at high magnification and provide strong evidence for all conclusions, presented in a concise manner. The study sheds light on the development and function of a glial cell type, the ensheathing glia, which we do not understand well so far. Their development and interactions with neurons and other glial cells, the role of polarity and the underlying molecular mechanisms, as well as the functional consequences of their disruption will be of high interest to the glial field in general. I look forward to seeing this study in print.

Minor comments:

Two little errors may need correction:

Line 88. "Likewise, the Excitatory amino acid transporter 2 ... is expressed by ensheathing glia and suppress sleep in the adult." is still not quite right.

Line 240. "Likewise, in when we ablated..." in should be removed.

We are thankful for this very positive judgement. We apologize for the two mistakes which we corrected.

[Redacted]

Reviewer #3 (Remarks to the Author):

The authors have done an outstanding job to respond in detail to the reviewers' comments.
The revised manuscript is acceptable for publication.

Thanks a lot for this encouraging comment!

References:

- Beckervordersandforth, R.M., Rickert, C., Altenhein, B., and Technau, G.M. (2008). Subtypes of glial cells in the *Drosophila* embryonic ventral nerve cord as related to lineage and gene expression. *Mechanisms of Development* *125*, 542–557.
- Ito, K., Urban, J., and Technau, G.M. (1995). Distribution, classification, and development of *Drosophila* glial cells in the late embryonic and early larval ventral nerve cord. *Roux's Archives of Developmental Biology* *204*, 284–307.
- Kremer, M.C., Jung, C., Batelli, S., Rubin, G.M., and Gaul, U. (2017). The glia of the adult *Drosophila* nervous system. *Glia* *65*, 606–638.
- Li, H.-H., Kroll, J.R., Lennox, S.M., Ogundeyi, O., Jeter, J., Depasquale, G., and Truman, J.W. (2014). A GAL4 driver resource for developmental and behavioral studies on the larval CNS of *Drosophila*. *Cell Rep* *8*, 897–908.
- Otto, N., Marelja, Z., Schoofs, A., Kranenburg, H., Bittern, J., Yildirim, K., Berh, D., Bethke, M., Thomas, S., Rode, S., et al. (2018). The sulfite oxidase *Shopper* controls neuronal activity by regulating glutamate homeostasis in *Drosophila* ensheathing glia. *Nat Commun* *9*, 3514.
- Peco, E., Davla, S., Camp, D., M Stacey, S., Landgraf, M., and van Meyel, D.J. (2016). *Drosophila* astrocytes cover specific territories of the CNS neuropil and are instructed to differentiate by Prospero, a key effector of Notch. *Development* *143*, 1170–1181.
- Steller, H. (2008). Regulation of apoptosis in *Drosophila*. *Cell Death Differ* *15*, 1132–1138.
- Vasudevan, D., and Ryoo, H.D. (2015). Regulation of Cell Death by IAPs and Their Antagonists. *Current Topics in Developmental Biology* *114*, 185–208.

Reviewers' Comments:

Reviewer #1:

Remarks to the Author:

I carefully assessed the comments of Reviewer 2 and the thorough and well-argued rebuttal of the authors.

Specificity of driver :

I agree with the authors that there is strong evidence that the driver 83E12-Gal4 is a reliable and specific marker for ensheathing glia (EG), most importantly supported by two publications (Li et al. Cell Reports, 2014 and Otto et al., Nat. Commun. 2018). The observation that some types of ensheathing glia also extend branches into the cortex (or the neuropil in some other brain areas) has been previously observed, and is not an indication that the driver is unfaithful. The arguments to provide further evidence through quantification of cell numbers in the rebuttal are convincing.

Dye diffusion experiments using glial ablation :

Here again, I agree with the authors. Killing glia is an entirely valid approach to determine barrier function comparing control and experimental animals. Any response of other glia to remove debris would still only be secondary and further enhance the loss of barrier function. In my view there would not be an alternative test to the described diffusion experiment.

Validity of PIP3 marker to assess polarity of EG :

The authors use two lipid sensors for PIP3 and PIP2 : "PH-PLC δ -mCherry is targeted by PIP2m whereas PH-AKT-GFP preferentially binds PIP3". The sensors are expressed in EG by the 83E12-Gal4 driver, thus polarity is indeed revealed only for EG not perineurial/sub-perineurial glia.

Usefulness of behavioral experiments :

In my view, these assays are insightful, as they show distinct changes in locomotion behavior/nervous system function. Disruption of barrier function could have been simply causing lethality, or other glial cells could have compensated. It also underscores the strength and breadth of genetic manipulations in this study as they reveal the role of this interesting glial cell type from the subcellular to the behavioral/functional level.

Reviewer #2:

[Redacted]

Reviewer #3:

Remarks to the Author:

The authors make a convincing rebuttal of the reviewer's criticisms. 1.I have no doubt that the driver they use is expressed in ensheathing glia
2.the dye injection experiment follows standard protocols and delivers interpretable results
3.The proliferation study, while leaving many details unanswered, also produces interpretable results; given that lineage of the ensheathing glia is only a side aspect of the study, I would not demand more experiments to address open questions concerning the origin of adult ensheathing glia.

Reviewer #4:

Remarks to the Author:

In this brief review of "Drosophila β Heavy-Spectrin is required in polarized ensheathing glia that form a diffusion-barrier around the neuropil" by Pogadalla et al., there are brief comments on 3 outstanding issues from the previous round of review. I believe the authors have sufficiently addressed previous reviewer #2's concerns.

Are the methods used to assess barrier function using dye diffusion sufficiently validated?

Yes. The Klambt lab was pioneering in its use of this approach to understand barrier functions in

Drosophila glia, and they have adapted the approach here by injecting dye into the contralateral brain lobe, an approach that cleverly overcomes several technical limitations and concerns related to the injection process itself. The authors did the appropriate control experiments, and for the purposes of this manuscript I agree with their contention in the rebuttal that this serves as useful and sufficient validation. The positioning and morphology of the ensheathing glia (EG) supports the idea they are the barrier for the transfer of substances to and from the neuropil. However, strictly speaking, ablation of the EG does not distinguish whether the EG actually form the barrier, or otherwise contribute to it indirectly. In a hypothetical example for illustrative purposes only, one might imagine how the presence of EG could help astrocytes to maintain the barrier function. So, I might suggest, at this early point in the manuscript prior to the analysis of beta-heavy spectrin, that the authors make a modest revision stating on line 300 "we conclude that the ensheathing glia, although they lack specialized occluding junctions, indeed contribute to a barrier function that possibly involves the extensive overlapping of ensheathing glia cell processes." I would also suggest that the descriptions of the Supplementary Movies 1 and 2 add some relevant descriptive details.

Is the 83E12-Gal4 driver specific for ensheathing glia, and could other cell types expressing the 83E12-Gal4 driver be a confounding factor for the main claims of this study?

Based on their figures of 83E12 expression, it is very specific to EG and EG/WG in L1, L2, and L3 larvae. Of course, the expression profile of a Gal4 driver can depend on the UAS reporter and the sensitivity of the detection method used to reveal its expression. In this instance, the Klambt lab used $P\{UAS-mCD8::GFP.L\}LL6$ (Bloomington 5130) and anti-GFP immunohistochemistry, a standard approach with high sensitivity. The functional effects on the barrier seen with 83E12 appear specific to ablation of ensheathing glia since, for example, 83E12 does not appear to drive sufficient rpr and hid expression in astrocytes to ablate them. The authors have carefully enumerated the 83E12-reporting cells in L3 larvae and this correlates well with what is known about EG from previous literature and their EM study in this manuscript. They also explain how some EG at the dorsal surface of the neuropil can be seen to wrap around neuronal cell bodies, something that Reviewer 2 might have interpreted as another cell type. Furthermore, they generated 83E12/repo split-Gal4 system to eliminate effects that might be caused by cells outside the nervous system (ie. midgut). Therefore, 83E12-Gal4 appears specific for EG in the CNS, and its expression in other cell types is not a confounding factor for the main claims of this study.

In the ablation paradigm, could inflammation be a confounding factor?

There is no evidence for this, and it seems an unusually vague critique of the work without further explanation or elaboration. The conclusions made by the authors are supported by their results. The defense of their conclusions in the rebuttal letter is rational and straightforward.

Reviewer #1 (Remarks to the Author):

I carefully assessed the comments of Reviewer 2 and the thorough and well-argued rebuttal of the authors.

Specificity of driver :

I agree with the authors that there is strong evidence that the driver 83E12-Gal4 is a reliable and specific marker for ensheathing glia (EG), most importantly supported by two publications (Li et al. Cell Reports, 2014 and Otto et al., Nat. Commun. 2018). The observation that some types of ensheathing glia also extend branches into the cortex (or the neuropil in some other brain areas) has been previously observed, and is not an indication that the driver is unfaithful. The arguments to provide further evidence through quantification of cell numbers in the rebuttal are convincing.

We are thankful for this positive judgement.

Dye diffusion experiments using glial ablation :

Here again, I agree with the authors. Killing glia is an entirely valid approach to determine barrier function comparing control and experimental animals. Any response of other glia to remove debris would still only be secondary and further enhance the loss of barrier function. In my view there would not be an alternative test to the described diffusion experiment.

We are thankful for this positive judgement.

Validity of PIP3 marker to assess polarity of EG :

The authors use two lipid sensors for PIP3 and PIP2 : "PH-PLC δ -mCherry is targeted by PIP2m whereas PH-AKT-GFP preferentially binds PIP3". The sensors are expressed in EG by the 83E12-Gal4 driver, thus polarity is indeed revealed only for EG not perineurial/sub-perineurial glia.

We are thankful for this positive judgement.

Usefulness of behavioral experiments :

In my view, these assays are insightful, as they show distinct changes in locomotion behavior/nervous system function. Disruption of barrier function could have been simply causing lethality, or other glial cells could have compensated. It also underscores the strength and breadth of genetic manipulations in this study as they reveal the role of this interesting glial cell type from the subcellular to the behavioral/functional level.

We are thankful for the very positive appreciation of our work and the support for our arguments.

[Redacted]

Reviewer #3 (Remarks to the Author):

The authors make a convincing rebuttal of the reviewer's criticisms.

1. I have no doubt that the driver they use is expressed in ensheathing glia
2. the dye injection experiment follows standard protocols and delivers interpretable results
3. The proliferation study, while leaving many details unanswered, also produces interpretable results; given that lineage of the ensheathing glia is only a side aspect of the study, I would not demand more experiments to address open questions concerning the origin of adult ensheathing glia.

We are very thankful for the very positive appreciation of our work.

Reviewer #4 (Remarks to the Author):

In this brief review of "Drosophila β Heavy-Spectrin is required in polarized ensheathing glia that form a diffusion-barrier around the neuropil" by Pogadalla et al., there are brief comments on 3 outstanding issues from the previous round of review. I believe the authors have sufficiently addressed previous reviewer #2's concerns.

We are thankful for the very positive appreciation of our work.

Are the methods used to assess barrier function using dye diffusion sufficiently validated?
Yes.

Thank you for the support.

The Klambt lab was pioneering in its use of this approach to understand barrier functions in Drosophila glia, and they have adapted the approach here by injecting dye into the contralateral brain lobe, an approach that cleverly overcomes several technical limitations and concerns related to the injection process itself. The authors did the appropriate control experiments, and for the purposes of this manuscript I agree with their contention in the rebuttal that this serves as useful and sufficient validation. The positioning and morphology of the ensheathing glia (EG) supports the idea they are the barrier for the transfer of substances to and from the neuropil. However, strictly speaking, ablation of the EG does not distinguish whether the EG actually form the barrier, or otherwise contribute to it indirectly.

We are again thankful for the positive view on our experiments and the very helpful and valid suggestion. We revised the text and clearly spelled out this possibility.

We changed the title of the respective section on page 12 which now reads
Ensheathing glial cells contribute to barrier formation around the neuropil

We also changed this in the discussion, where we added on page 20:

A likely role of ensheathing glial cells is to establish a diffusion barrier around the neuropil, as it was demonstrated by our dye injection experiments in ensheathing glia ablated larvae. However, we cannot exclude the possibility that ensheathing glia instruct neighboring cells such as the cortex glia or astrocyte-like glia to form a barrier around the neuropil.

In a hypothetical example for illustrative purposes only, one might imagine how the presence of EG could help astrocytes to maintain the barrier function. So, I might suggest, at this early point in the manuscript prior to the analysis of beta-heavy spectrin, that the authors make a modest revision stating on line 300 “we conclude that the ensheathing glia, although they lack specialized occluding junctions, indeed contribute to a barrier function that possibly involves the extensive overlapping of ensheathing glia cell processes.”

We are very thankful for this suggestion and changed the paper accordingly on page 13:
Thus, we conclude that the ensheathing glia, although they lack specialized occluding junctions, contribute to a barrier function that possibly involves the extensive overlap noted for ensheathing glia cell processes (Figure 2).

I would also suggest that the descriptions of the Supplementary Movies 1 and 2 add some relevant descriptive details.

We are sorry for this lack of information. We provide a better description of the supplementary Movies 1 and 2.

Is the 83E12-Gal4 driver specific for ensheathing glia, and could other cell types expressing the 83E12-Gal4 driver be a confounding factor for the main claims of this study?
Based on their figures of 83E12 expression, it is very specific to EG and EG/WG in L1, L2, and L3 larvae. Of course, the expression profile of a Gal4 driver can depend on the UAS reporter and the sensitivity of the detection method used to reveal its expression. In this instance, the Klambt lab used P{UAS-mCD8::GFP.L}LL6 (Bloomington 5130) and anti-GFP immunohistochemistry, a standard approach with high sensitivity. The functional effects on the barrier seen with 83E12 appear specific to ablation of ensheathing glia since, for example, 83E12 does not appear to drive sufficient *rpr* and *hid* expression in astrocytes to ablate them. The authors have carefully enumerated the 83E12-reporting cells in L3 larvae and this correlates well with what is known about EG from previous literature and their EM study in this manuscript. They also explain how some EG a at the dorsal surface of the neuropil can be seen to wrap around neuronal cell bodies, something that Reviewer 2 might have interpreted as another cell type. Furthermore, they generated 83E12/repo split-Gal4 system to eliminate effects that might be caused by cells outside the nervous system (ie. midgut). Therefore, 83E12-Gal4 appears specific for EG in the CNS, and its expression in other cell types is not a confounding factor for the main claims of this study.

We are thankful for this positive judgement.

In the ablation paradigm, could inflammation be a confounding factor?
There is no evidence for this, and it seems an unusually vague critique of the work without further explanation or elaboration. The conclusions made by the authors are supported by their results. The defense of their conclusions in the rebuttal letter is rational and straightforward.

We are thankful for this positive judgement.